# PCGF6 controls neuroectoderm specification of human pluripotent stem cells by activating SOX2 expression

Xianchun Lan[1], Song Ding [1], Tianzhe Zhang[1], Ying Yi[1], Conghui Li[2], Wenwen Jin[1], Jian Chen[3], Kaiwei Liang[2], Hengbin Wang [4] & Wei Jiang [1,5,6] ✉

Polycomb group (PcG) proteins are known to repress developmental genes during embryonic development and tissue homeostasis. Here, we report that PCGF6 controls neuroectoderm specification of human pluripotent stem cells (PSCs) by activating *SOX2* gene. Human PSCs with PCGF6 depletion display impaired neuroectoderm differentiation coupled with increased mesendoderm outcomes. Transcriptome analysis reveals that de-repression of the WNT/β-catenin signaling pathway is responsible for the differentiation of PSC toward the mesendodermal lineage. Interestingly, PCGF6 and MYC directly interact and co-occupy a distal regulatory element of SOX2 to activate SOX2 expression, which likely accounts for the regulation in neuroectoderm differentiation. Supporting this notion, genomic deletion of the SOX2-regulatory element phenocopies the impaired neuroectoderm differentiation, while overexpressing SOX2 rescues the neuroectoderm phenotype caused by PCGF6-depletion. Together, our study reveals that PCGF6 can function as lineage switcher between mesendoderm and neuroectoderm in human PSCs by both suppression and activation mechanisms.

Polycomb group (PcG) protein-mediated transcription repression mechanisms have been studied extensively. PcG proteins, as repressors of developmental genes, were originally identified in Drosophila melanogaster and described as being able to control segmentation by maintaining temporal and spatial repression of Hox gene during early embryogenesis[1]. Subsequently, the PcG proteins have been shown to be necessary for regulation of developmental genes through chromatin modifications[2,3].

Based on the composition as well as enzymatic activity toward specific histone residues, PcG complexes have been broadly classified into two major complexes, named Polycomb repressive complex 1 (PRC1) and Polycomb repressive complex 2 (PRC2)[4,5]. In mammals, PRC1 consists of five CBX homologs (CBX2/4/6/7/8), six PCGF family members (PCGF1-6), three PHC family members (PHC1-3), and two ubiquitin ligases (RING1A/1B) which catalyze the ubiquitylation of lysine 119 of histone H2A (H2AK119ub) and promote chromatin compaction[6]. PRC2 consists the core components EZH2 or its closely related homolog EZH1, and EED and SUZ12, which catalyze the methylation of lysine 27 of histone H3 (H3K27me)[7].

The establishment of Polycomb-dependent transcriptional repression requires complicated interactions between PRC1 and PRC2[8]. Previous studies revealed that PRC2 was first recruited into the Polycomb domain and catalyzed the establishment of H3K27me3, which was then recognized by the CBX proteins and recruited PRC1 to

[1]Department of Biological Repositories, Frontier Science Center for Immunology and Metabolism, Medical Research Institute, RNA Institute, Zhongnan Hospital of Wuhan University, Wuhan University, Wuhan 430071, China. [2]Department of Pathophysiology, School of Basic Medical Sciences, Wuhan University, Wuhan 430071, China. [3]Chinese Institute for Brain Research (Beijing), Research Unit of Medical Neurobiology, Chinese Academy of Medical Sciences, 102206 Beijing, China. [4]Department of Internal Medicine, Division of Hematology, Oncology, and Palliative Care, Massey Cancer Center, Virginia Commonwealth University, Richmond, VA 23298, USA. [5]Human Genetics Resource Preservation Center of Wuhan University, Wuhan, China. [6]Hubei Provincial Key Laboratory of Developmentally Originated Disease, Wuhan, China. ✉e-mail: jiangw.mri@whu.edu.cn

deposit H2AK119ub[4,7]. Two main mechanisms have been ascribed to Polycomb-mediated gene repression so far. First, PRC1 complexes have the ability to establish a compact chromatin state and create long-range interactions between Polycomb target sites, which could maintain the repression of target genes[9,10]. Second, the H2AK119ub deposition catalyzed by PRC1 as a central hub that contributes to the stabilization of PRC1 complexes and PcG-mediated transcriptional repression[11,12]. Both mechanisms can function independently, but also complement each other[13].

Recently, several reports indicated that the regulation of PRC1 is far more complicated than previously thought. The discovery of variant PcG complexes suggests that there are six distinct PRC1 sub-complexes, which are determined by different RING-PCGF heterodimers and referred to as noncanonical PRC1 (ncPRC1, also named PRC1.1-1.6)[14,15]. The six RING-PCGF heterodimer combinations form multiple complexes through association with RYBP, which prevents the binding of other PRC1 components, such as CBX proteins. These ncPRC1s could be recruited to the Polycomb domain without PRC2. Meanwhile, they were found to have greater ubiquitinating activity toward H2AK119 than the canonical PRC1[14]. The overlap between different ncPRC1 is minimal, involving their accessory subunit composition, genomic localization, and functional characteristics[15].

In line with this, individual PCGF family members have been studied during last decade, which highlighted the different roles of PCGFs in mammalian development. For instance, PCGF1, also known as NSPc1, terminates self-renewal and primes differentiation program in hematopoietic progenitor cells by repressing HoxA cluster genes[16]. PCGF2, known as MEL18, specifies mesoderm specification in mouse embryonic stem cells (ESCs) via negatively regulating pluripotency genes and positively regulating key mesoderm transcription factors[17]. PCGF3/5 initiates the recruitment of both PRC1 and PRC2 in response to *Xist* RNA expression in X chromosome inactivation[18]. In addition, PCGF5 has been shown to cooperate with the Auts2 transcription factor and function as a repressor for SMAD2/TGF-β signaling pathway in neural differentiation[19,20].

Polycomb group ring finger 6 (PCGF6), also known as MBLR and RNF134, has been identified as a member of noncanonical PRC1.6 complex, together with the E2F6 transcription factor[21]. RING finger domain and WD40-associated ubiquitin-like (RAWUL) domain exist in PCGF6, yet only RING finger domain enables PCGF6 to directly associate with the RING1A/B[22].

PCGF6 is highly expressed in multiple tissues and cell types during mouse early embryonic development. Mice with homozygous deletion of PCGF6 are viable and fertile but are not born at the normal Mendelian ratio[23,24], suggesting an embryonic developmental abnormality. Indeed, embryonic death was continuously detected in *Pcgf6*[−/−] homozygous knockout embryos from the blastocyst stage through post-implantation development, while surviving *Pcgf6*[−/−] embryos showed a growth retardation phenotype[23,24]. Moreover, knockdown of *Pcgf6* in mouse ESCs decreases the expression of core pluripotency factors, and complete loss of *Pcgf6* in mouse ESCs leads to robust de-repression of germ cell-related genes[23,25]. These studies suggest that PCGF6 is important for mouse early development via PRC1.6 integrity. However, whether PCGF6 can function independent on PRC1, and whether PCGF6 modulates pluripotency and lineage specification and contributes to early development in human, remain poorly understood.

Here, we established PCGF6-knockout (PCGF6-KO) human pluripotent stem cells (PSCs). PCGF6 deficiency does not affect the key features of pluripotency but does change the gene expression balance between different lineages in human PSCs. Using multiple differentiation assays we elucidate the divergent roles of PCGF6 in distinct lineage specification. At last, we dissect the mechanisms for PCGF6 in balancing early lineage differentiation of human PSCs.

## Results

### PCGF6 is not necessary for human PSC maintenance

To understand the function of PCGF6 in human PSCs, we knocked out the *PCGF6* gene in human induced PSC line PGP1 via CRISPR-Cas9 genome editing by targeting the first exon. We examined 24 colonies in total and successfully obtained two knockout clones with different genotype (KO#1, 178 bp deletion at both alleles; KO#2, 203 bp deletion at both alleles) (Supplementary Fig. 1a). PCGF6-KO was validated by genomic PCR, qRT-PCR and Western blot (Supplementary Fig. 1b, c and Fig. 1a). PCGF6-KO human PSCs are still able to form colonies and display positive alkaline phosphatase staining, but the colonies are flat, loose, and morphologically distinctive (Fig. 1b). PCGF6-KO did not affect the cell cycle, proliferation and maintenance of the PSCs although the colony size became significantly smaller (Fig. 1c, d and Supplementary Fig. 1d). Furthermore, the qRT-PCR analysis and immunofluorescence staining showed that knockout of PCGF6 does not significantly impact the RNA and protein levels of the core pluripotent factors OCT4 and NANOG (Fig. 1e, f). Western blot analysis also demonstrated the comparable expression levels of OCT4 and NANOG between PCGF6-KO and wild-type (WT) cells (Supplementary Fig. 1e). In addition, PCGF6-KO did not affect the protein levels of other components of PRC1.6 (Supplementary Fig. 1f). We further checked the stability of PRC1.6 complex by co-immunoprecipitation with anti-RING1B antibody in both wildtype and PCGF6-depleted human PSCs. While RING1B readily co-immunoprecipitated with endogenous L3MBTL2, MAX, RYBP, E2F6, MGA and EHMT2 in wild-type human PSCs, the interaction between RING1B and L3MBTL2, MAX, E2F6, MGA, EHMT2 were severely impaired in PCGF6-KO cells. We also noticed that the depletion of PCGF6 did not disrupt the interaction between RYBP and RING1B (Fig. 1g). Taken together, these data indicate that PCGF6 is dispensable for maintenance of human PSCs but plays an essential role in PRC1. 6 complex integrity.

### PCGF6 depletion causes aberrant gene expression and spontaneous differentiation

To better understand the molecular function of PCGF6 in human PSCs, we performed RNA sequencing (RNA-seq) experiment using PCGF6-KO and wild-type human PSCs. We found that PCGF6 disruption did not affect the expression of the key pluripotency genes (Supplementary Fig. 1g), which is consistent with our previous results (Fig. 1e, f). Besides, loss of PCGF6 did not appear to appreciably affect the mRNA levels of the other components of PRC complexes (Supplementary Fig. 1h). With the cutoff fold-change >1.5 and $p < 0.05$, we identified 3442 differentially expressed genes in PCGF6-KO human PSCs compared with wild-type (Fig. 2a and Supplementary Data 1), most of which (2188/3442) were upregulated, suggesting PCGF6 mainly functions as transcriptional repressor as expected. Many of the significantly upregulated genes upon PCGF6-KO were mesodermal and endodermal genes, associated with developmental processes including skeletal system development, organ morphogenesis, heart development, and cell fate commitment (Fig. 2b, c). Meanwhile, the downregulated genes were enriched in synapse, central nervous system development and neuronal cell body (Fig. 2b, c). These results promote us to ask whether the absence of PCGF6 has any effect on the capacity of lineage differentiation of human PSCs.

In general, embryoid body (EB) could mimic early embryonic development in vitro and were often utilized for testing the capacity of human PSC differentiation[26]. Thus, we performed EB assay by hanging-drop followed with suspension culture (Fig. 2d, upper panel). Through examining EB morphology, we found that PCGF6-KO human PSCs formed EBs normally, with similar size to those of the wild-type PSCs (Fig. 2d, bottom panel). The qRT-PCR analysis of 9-day EBs revealed that the expression levels of pluripotency markers (*OCT4* and *NANOG*) were higher, accompanied by decreased neuroectoderm markers

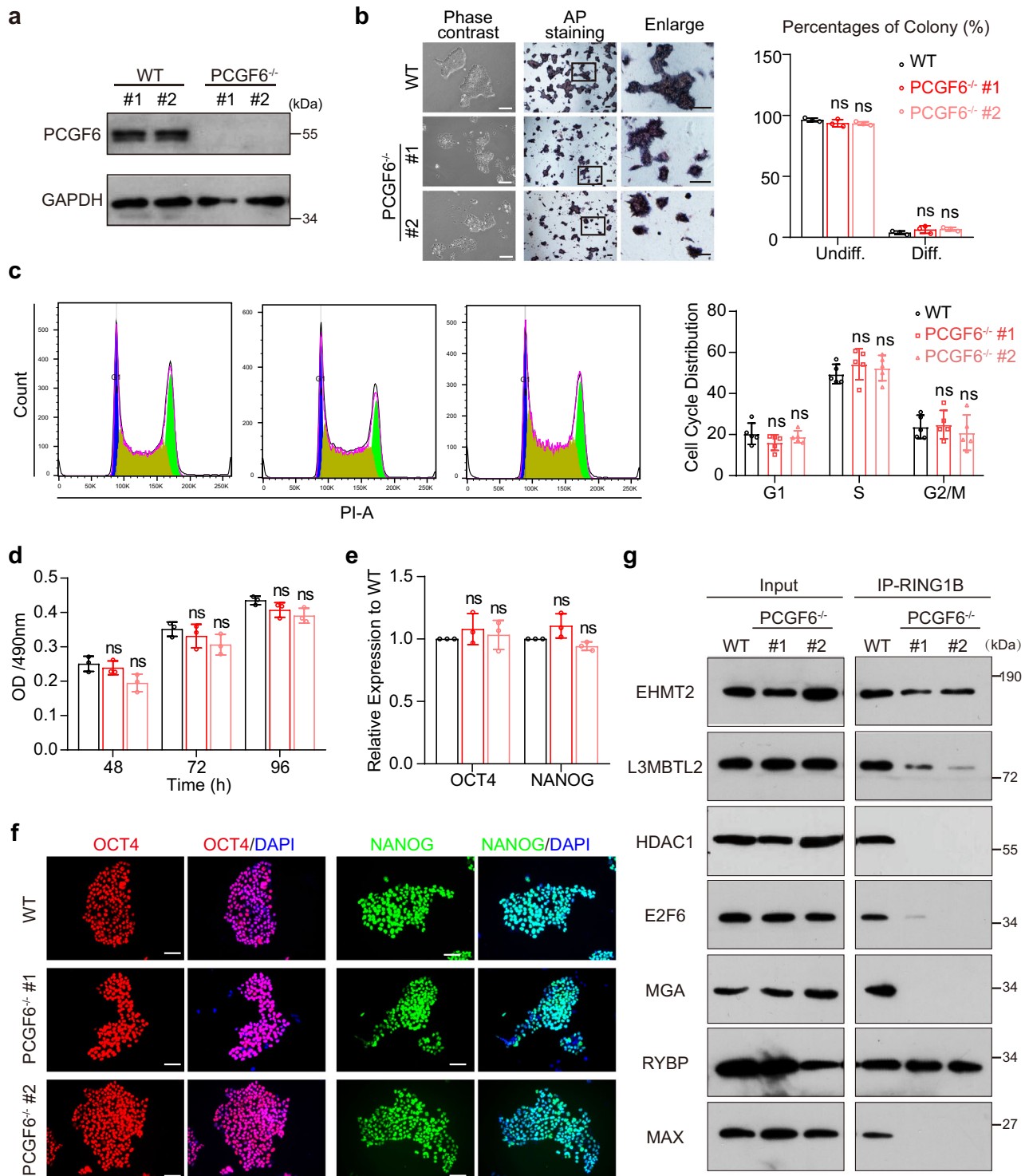

**Fig. 1 | PCGF6 is dispensable for maintenance of human PSCs. a** PCGF6 protein level in WT and PCGF6-KO human PSCs; GAPDH was used as loading control (*n* = 3). **b** Left: representative images showing the morphology (left panels) and alkaline phosphatase activity (right panels) of WT and PCGF6-KO human PSCs. Scale bars, 200 μm. Right: bar graph showing the mean percentages of differentiated (AP-negative) and undifferentiated (AP-positive) colonies in WT and PCGF6-KO human PSCs (*n* = 3). ns (not significant), *p* > 0.05 compared with WT. **c** Cell cycle analysis using PI staining in WT and PCGF6-KO human PSCs. The populations of different phases were calculated in FlowJo (v10.4.0) (*n* = 5). **d** MTT analysis of proliferation rate in WT and PCGF6-KO human PSCs (*n* = 3). **e** Relative expression of *OCT4* and *NANOG* in the WT and PCGF6-KO human

PSCs. The level of gene expression in the WT human PSCs is set as 1 (*n* = 3). **f** WT and PCGF6-KO human PSCs were immunofluorescently stained for OCT4 or NANOG expression, and nuclei were counterstained with DAPI (*n* = 3). Scale bars, 200 μm. **g** Immunoprecipitation of PRC1.6 components (EHMT2, L3MBTL2, HDAC1, E2F6, MGA, RYBP, MAX) followed by western blot with an antibody against RING1B in WT and PCGF6-KO cells. 10% of the total cell lysate used for each immunoprecipitation was loaded as input (*n* = 3). Each point represents a biological replicate. Data are presented as the mean ± SD. Statistical significance was determined using the unpaired, two-tailed *t*-test in **b**–**e** (ns not significant, *p* < 0.05, **p* < 0.01, ***p* < 0.001). Exact *p* values and statistical parameters are provided as a Source Data file.

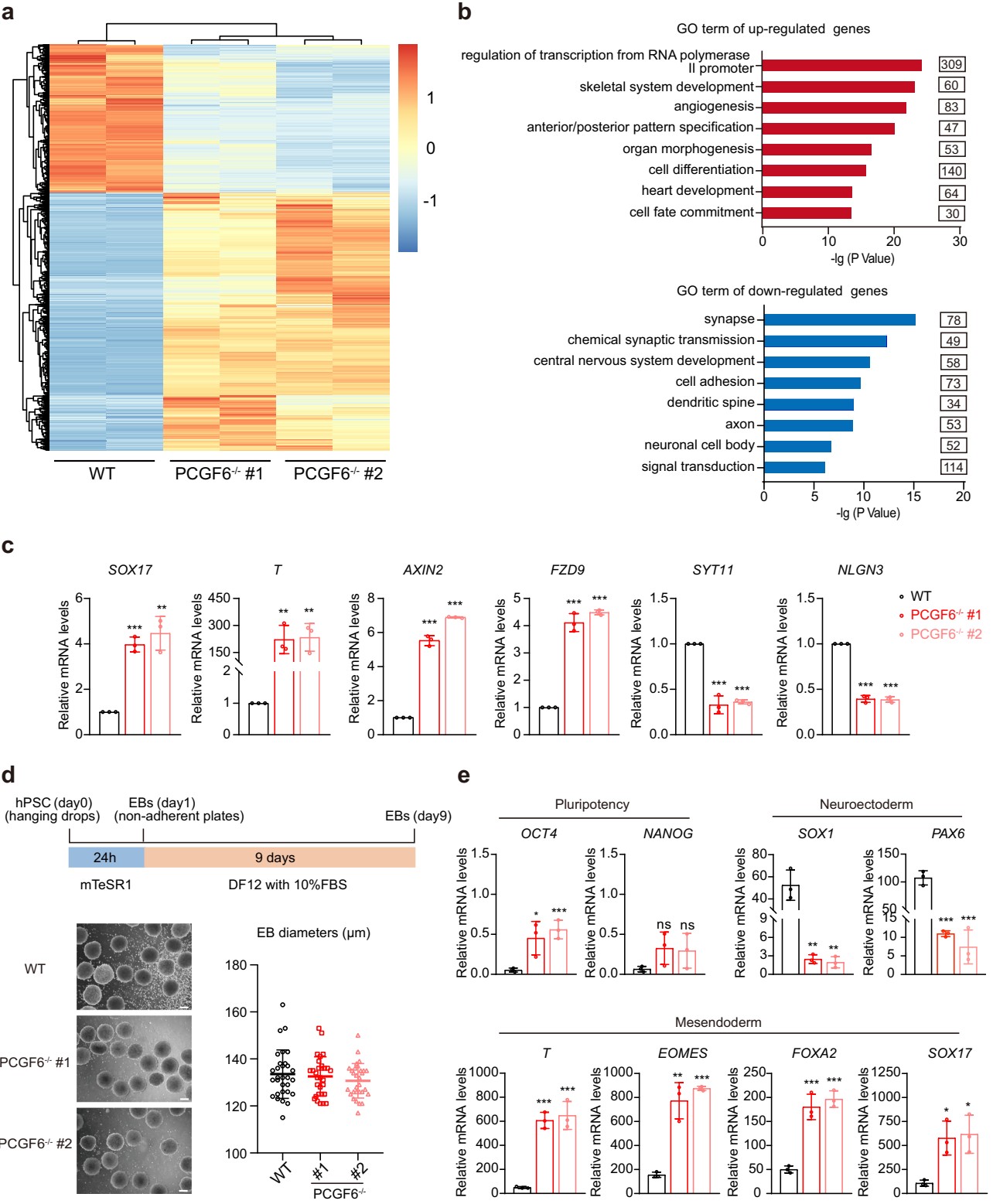

**Fig. 2 | Depletion of PCGF6 results in aberrant gene expression and the imbalance of differentiation. a** The heatmap showed the differentially expressed genes (DEGs) in WT and PCGF6-KO PSCs. **b** Gene Ontology analysis of upregulated (top panel) and downregulated (bottom panel) genes in response to PCGF6-knockout. **c** qRT-PCR was used to measure the relative mRNA expression of differentially expressed genes (*SOX17* and *T* for mesendoderm, *AXIN2 and FZD9* for WNT signaling pathway, *SYT11* and *NLGN3* for neuroectoderm) in WT and PCGF6-KO human PSCs. The level of gene expression in the WT human PSCs is set as 1 (*n* = 3). **d** Top panel: schematic representation of Embryoid body (EB) differentiation of human PSCs over 9 days. Bottom panel:

bright-field images of WT and PCGF6-KO EBs at days 9 in suspension culture (left), Scale bars, 100 μm; scatter graph displayed the mean diameter of 30 random EBs from WT and PCGF6-KO cell lines (*n* = 30) (right). **e** qRT-PCR was used to measure the relative mRNA expression levels of pluripotency markers (*OCT4* and *NANOG*) and lineage-specific markers (endoderm, mesoderm and ectoderm) in WT and PCGF6-KO EBs on day 9 (*n* = 3). Each point represents a biological replicate. Data are presented as the mean ± SD. Statistical significance was determined using the unpaired, two-tailed *t*-test in **c**–**e** (ns not significant, *$p < 0.05$, **$p < 0.01$, ***$p < 0.001$). Exact *p* values and statistical parameters are provided as a Source Data file.

(*SOX1* and *PAX6*) but increased mesendoderm markers (*T, EOMES, FOXA2* and *SOX17*) in PCGF6-KO EBs compared with wild-type (Fig. 2e).

Collectively, these data suggested that PCGF6-KO favors a mesendoderm trajectory and blocks the development of neuroectoderm, indicating a critical role of PCGF6 in lineage fate decision.

## PCGF6 directly represses WNT/β-catenin signaling gene expression via the PRC1-dependent repression in human PSCs

We further performed directed differentiation of wild-type and PCGF6-KO human PSCs into definitive endoderm lineage[27,28] and found that compared to wild-type cultures, a robust population of CXCR4-positive cells increased in the PCGF6-KO cultures (Supplementary Fig. 2a–c), which also indicated that knockout of PCGF6 promoted mesendodermal differentiation. Furthermore, we performed lineage differentiation toward endodermal pancreatic progenitors[29] and mesodermal cardiomyocytes[30] using PCGF6-KO and wild-type PSCs. The result from pancreatic differentiation showed that the induction of key pancreatic lineage markers (*PDX1* and *NKX6-1*) was much higher in PCGF6-KO cells (Supplementary Fig. 2d, e). Similarly, cardiomyocytes differentiated from PCGF6-KO human PSCs exhibited increased *T, NKX2.5* and *GATA4* expression (Supplementary Fig. 2f, g). These results together confirmed the inhibitory role of PCGF6 in mesendodermal fate determination. To dissect the underlying mechanism of skewed lineage differentiation upon deleting PCGF6, we further analyzed the impacted signal pathways. Gene set enrichment analysis and KEGG pathway analysis based on transcriptome highlighted that the expression of WNT/β-catenin signaling genes were generally upregulated in PCGF6-KO human PSCs than in wild-type (Fig. 3a and Supplementary Fig. 2h). WNT/β-catenin signaling pathway is well-known to promote mesendoderm differentiation of human PSCs[31,32], thus we suspected PCGF6 modulated WNT activity for mesendoderm differentiation. We performed PCGF6 ChIP-seq experiment and found a significant portion of WNT signaling genes appeared in PCGF6-binding (identified by ChIP-seq) and PCGF6-repressing (identified by RNA-seq) genes (Fig. 3b). More importantly, deletion of PCGF6 resulted in a significant increase in the amount of Non-phosphorylated (Active form) β-Catenin in human PSCs (Fig. 3c). In addition, overexpression of PCGF6 in 293T cells repressed the TCF reporter activity (Fig. 3d), which contained seven copies of the Tcf/lef binding site[33]. These data together support that PCGF6-KO resulted in abnormal activation of WNT/β-catenin signaling pathway which contributed to the increased expression of mesendodermal lineage genes in PSCs (Fig. 2c) and biased mesendodermal differentiation observed in EB and direct differentiation assays (Fig. 2e and Supplementary Fig. 2b–g).

Given that many WNT genes were upregulated upon PCGF6 deletion, we were wondering whether PCGF6 directly repressed the expression of these genes. Previous reports identified PCGF6 as component of PRC1.6, which epigenetically repressed the transcription of key developmental genes[14,34,35]. Therefore, we compared the enrichment of PCGF6, RYBP, RING1B, MAX, E2F6 and H2AK119ub on PCGF6 negatively-regulated binding sites by ChIP-seq datasets in human PSCs and found a significant overlap (Fig. 3e). A de novo sequence motif analysis of the PCGF6 binding sites in upregulated genes revealed centrally enriched motifs that matched in vitro recognition sequences for MAX (the E-Box, CACGTG)[36] and E2F6 (GCGGGAA)[37] (Fig. 3f), the major components of PRC1.6. ChIP-seq data indicated that PCGF6, RYBP, RING1B, E2F6, MAX and H2AK119ub were indeed specifically recruited to the promoters of these WNT signaling genes, such as FZD9 and AXIN2 (Fig. 3g). To further investigate the effect of PCGF6 deletion on the deposition of H2AK119ub, we performed ChIP-seq with anti-H2AK119ub antibody in both wild-type and PCGF6-KO human PSCs. Our data showed that loss of PCGF6 significantly decreased the level of H2AK119ub near the promoters of these WNT signaling genes in human PSCs, indicating PCGF6 directly represses WNT signaling genes expression via

PRC1.6 complex (Fig. 3e, g). In addition, we performed ChIP-seq experiments using anti-MAX and anti-E2F6 antibodies in wild-type and PCGF6-depleted human PSCs. There are 398 genes directly repressed by PCGF6 and targeted by PRC1.6 (overlapped with targets of RING1B, RYBP, E2F6 and MAX), many of which are within the WNT signaling pathway (such as *FZD9, WIN5A, WIN11*) (Supplementary Fig. 2i, j). ChIP-qPCR results showed that the enrichment of RING1B, EHMT2 and HDAC1 on genes in the WNT signaling pathway is indeed significantly reduced in the absence of PCGF6 (Supplementary Fig. 2k). These results suggest that PCGF6 directly represses WNT signaling genes through the PRC1-dependent manner. However, the signal of the MAX and E2F6 peaks in PCGF6-KO group are almost unchanged or even slightly increased (Supplementary Fig. 2j), which is consistent with a previous report that PCGF6-KO did not affect the overall genomic binding positions of the PRC1.6 in 293T and mouse ESCs[38].

Taken together, these data revealed that PCGF6 deletion induces skewed differentiation toward mesendoderm in human PSCs mainly through the PRC1-dependent repression of WNT/β-catenin signaling genes.

## PCGF6 regulates early neuroectoderm differentiation by positively regulating SOX2

Since PCGF6 depletion resulted in significant downregulation of a portion of PCGF6 target genes which statistically enriched for neuro-developmental categories (Figs. 2b and 4a), and PCGF6 was upregulated during neuroectoderm differentiation (Fig. 4b), we hypothesized PCGF6 may play a distinct role in neuroectodermal fate determination. Of note, *SOX2* and its target genes appeared in the group which was positively regulated by PCGF6 and bound by PCGF6 (Fig. 4c). To determine whether SOX2 was the direct target of PCGF6, we further examined the expression at RNA and protein levels, which confirmed that PCGF6 depletion caused significant decrease of SOX2 expression (Fig. 4d, e). SOX2 has been identified as a determining factor of neuroectoderm lineage[39–41]. Thus, to investigate the effects of PCGF6 on the neuroectodermal differentiation potentials of human PSCs, we subjected the PCGF6-KO and wild-type human PSCs to a well-developed neuroectoderm differentiation system[42], which allowed the generation of highly homogeneous neural progenitor cells from human PSCs. To monitor the dynamic changes of gene expression during neuroectoderm differentiation of human PSCs upon deletion of PCGF6, we performed qRT-PCR experiments at differentiation day 0, 3, 5 and 7 in both wild-type and PCGF6-KO human PSCs (Supplementary Fig. 3a). After 7 days' neuroectoderm induction, OCT4 and NANOG were downregulated in both wild-type and PCGF6-KO cells, although the downregulation was less efficient in PCGF6-KO group (Supplementary Fig. 3b, c). More importantly, the neuroectoderm markers (*SOX1, PAX6*) were significantly upregulated in wild-type cells, while fewer neuroectoderm markers were present in PCGF6-KO cells, indicating that early neuroectoderm differentiation was significantly blocked (Fig. 4f). Immunofluorescence staining also indicated fewer cells expressing SOX1 or PAX6 in PCGF6-KO cells compared to wild-type cells (Fig. 4g). Importantly, these neuroectoderm differentiation change can be rescued by re-expression of PCGF6 (PCGF6[−/−] + PCGF6) (Supplementary Fig. 3d–g), indicating that the observed neuroectoderm differentiation change is indeed caused by PCGF6 deficiency. In addition, we performed neuronal differentiation using PCGF6-KO and wild-type PSCs and hardly observed TUJ1-positive and SOX1-positive cells only in PCGF6-KO group (Supplementary Fig. 3h, i), further confirming the essential role of PCGF6 in neural lineage differentiation.

To investigate whether SOX2 is the functional target of PCGF6 in neuroectoderm differentiation and the ectopic expression of SOX2 could rescue the neuroectoderm defect phenotype in PCGF6-KO cells,

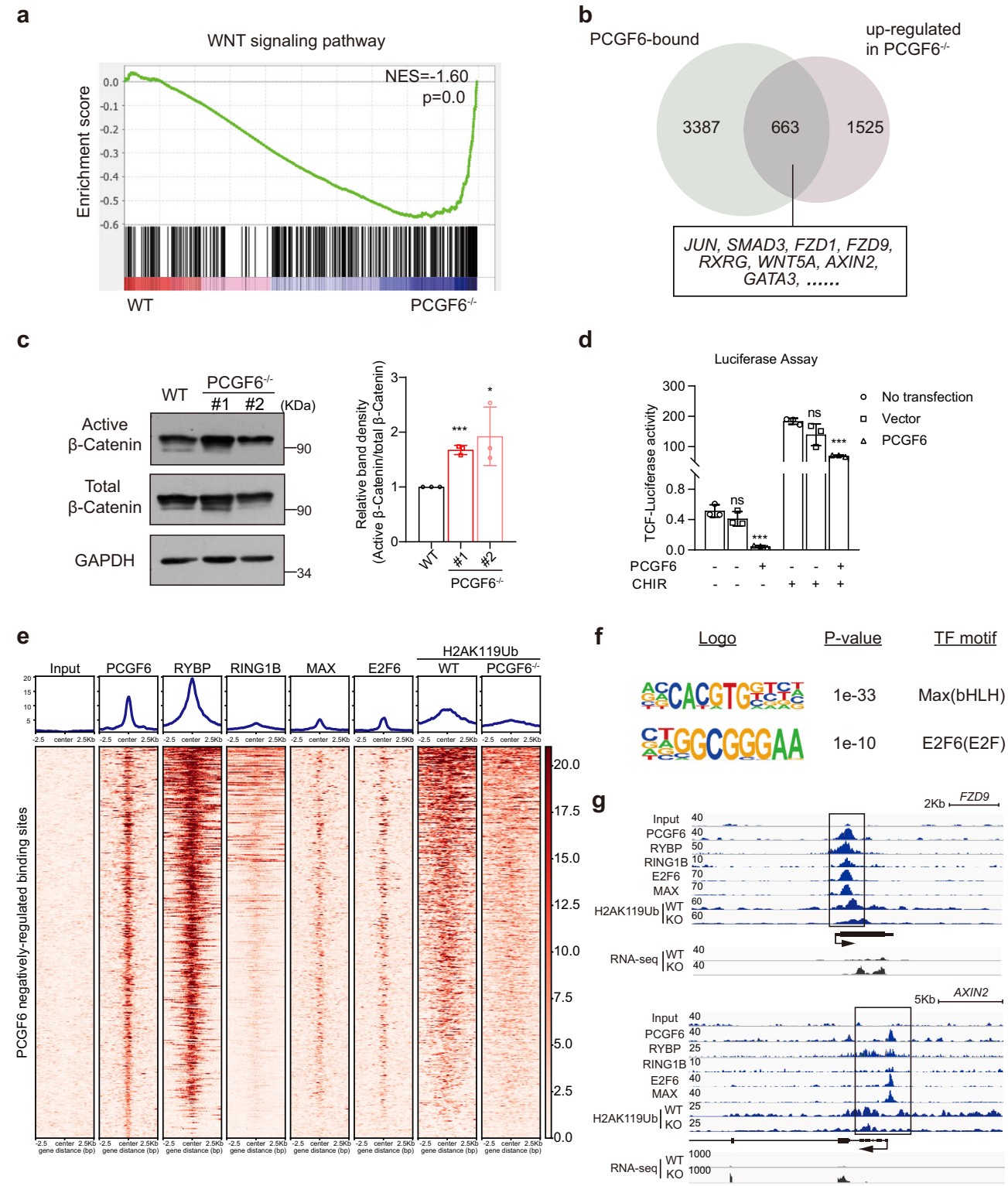

we stably expressed SOX2 in the PCGF6-KO human PSCs via lentiviral transduction (PCGF6$^{-/-}$ + SOX2) (Fig. 4e). We then induced neuroectoderm differentiation of PCGF6$^{-/-}$ + SOX2 human PSCs, and qRT-PCR analysis at different time-points of differentiation demonstrated that the defect in expression of neuroectoderm markers due to the loss of PCGF6 was partially rescued in SOX2-transfected PCGF6-KO human PSCs (Fig. 4f). In addition, immunofluorescence staining of neural progenitor markers SOX1 and PAX6 also supported that SOX2 restoration rescued the neural differentiation defect due to PCGF6 deletion (Fig. 4g).

Taken together, these data demonstrated that the knockout of PCGF6 caused the destruction of the neuroectodermal differentiation, mainly through activating SOX2 expression.

## PCGF6 interacts with MYC and contributes to transcriptional activation of SOX2

To reveal the mechanism by which PCGF6 activates SOX2 expression, we looked into our PCGF6 ChIP-seq data and found PCGF6 could directly bind to a distal region around 10 kb downstream of *SOX2* gene body (Fig. 5a), which was further validated by ChIP-qPCR analysis

**Fig. 3 | Depletion of PCGF6 in human PSC activates WNT signaling pathway.**
**a** GSEA profile of WNT signaling pathway genes in WT and PCGF6-KO PSCs.
**b** Venn diagram shows the overlap of PCGF6-binding genes in WT cells and upregulated genes in PCGF6-KO cells, revealing WNT signaling pathway genes as potential direct targets of PCGF6. **c** Protein levels of active β-catenin and total β-catenin were evaluated in WT and PCGF6-KO human PSCs (left panel); quantitative analysis of active β-catenin over total β-catenin was performed (right panel). *$p < 0.05$; ***$p < 0.001$ compared with WT ($n = 3$). **d** the 7xTCF luciferase reporter assay showed that PCGF6 inhibited WNT signaling pathway ($n = 3$). Left: ***$p < 0.01$ compared with PCGF6−/CHIR−; Right: ***$p < 0.01$ compared with PCGF6−/CHIR+. **e** The heatmap showing the enrichment of PCGF6, RYBP, RING1B, MAX, E2F6 and H2AK119ub (WT and PCGF6-KO) at upregulated PCGF6 target genes in human PSCs. ChIP-seq data of RYBP and RING1B were

obtained from GEO database: GSM2805870 and GSM2805868, respectively. **f** Enriched sequence motif of the PCGF6 binding regions in upregulated genes. The homer module findMotifsGenome.pl were run to obtain enriched sequence. The statistical significance ($p$ value) and the transcription factors (TF) that bind to the motif are shown. **g** Bedgraph for PCGF6, RYBP, RING1B, E2F6, MAX and H2AK119ub (WT and PCGF6-KO) ChIP-seq data at two WNT genes *FZD9* and *AXIN2* loci. The *x* axis corresponds to genomic locations with the scale indicated at the top of panel. The *y* axis corresponds to ChIP-seq or RNA-seq signal intensity. Each point represents a biological replicate. Data are presented as the mean ± SD. Statistical significance was determined using the unpaired, two-tailed *t*-test in **c**, **d** (ns not significant, *$p < 0.05$, **$p < 0.01$, ***$p < 0.001$). Exact *p* values and statistical parameters are provided as a Source Data file.

(Fig. 5b). This region is enriched with active histone markers such as H3K4me1, H3K4me2, H3K4me3 and H4K8ac (Fig. 5a), indicating a regulatory element of SOX2 (SRE). We also surveyed the binding of individual PRC1.6 component on SRE, by investigating the publicly available ChIP-seq data from the ENCODE/GEO database. The result showed that although some components of PRC1.6 (such as MAX, E2F6 and RYBP) occupied the SRE as well, RNF2/RING1B, the catalytic subunit for H2AK119 ubiquitination, is not enriched on SRE (Supplementary Fig. 4a–c). In addition, the public ChIP-seq data of HDAC2 and our ChIP-qPCR results of EHMT2 and HDAC1 all indicated that these three repressive histone modifiers are not enriched on SRE (Supplementary Fig. 4a, b). Taken together, while PCGF6 is not the only PRC1.6 subunit recruited to the SRE site, a functional PRC1.6 may not form on SRE site. We then generated SRE-KO human PSCs by deleting this genomic region of SRE using CRISPR/Cas9 (Supplementary Fig. 4d, e). As we expected, SRE-KO human PSCs displayed similar colony morphology to PCGF6-KO human PSCs (Fig. 5c). Furthermore, qRT-PCR and western blot analyses demonstrated that the deletion of SRE indeed significantly reduced the expression of SOX2 in human PSCs, both at RNA and protein levels (Fig. 5d, e). To assess the ability of SRE-KO human PSCs to generate neural progenitor cells, day 3 and day 7 cells were collected and stained for the expression of SOX1 and PAX6, respectively. Immunofluorescence staining revealed SRE-KO group exhibited fewer SOX1+ and PAX6+ cells compared with wild-type group, similar to PCGF6-KO group (Fig. 5f, g). Thus, our results suggested that PCGF6 can directly bind to the regulatory element downstream of SOX2 and affect neuroectoderm lineage differentiation by transcriptionally activating the expression of SOX2.

Since our ChIP-seq data demonstrated that PCGF6-bound genes did not completely involve in H2AK119ub enrichment (Supplementary Fig. 4f), it prompted us to further investigate which factor PCGF6 interacts with to activate the transcription of SOX2 at SRE locus. We surveyed a panel of ChIP-seq dataset from public database (ENCODE human ESC ChIP-seq) and intriguingly, MYC, a well-known transcriptional factor, occupied the SRE region (Fig. 5a). We further determined the protein interaction between PCGF6 and MYC. The co-immunoprecipitation assay showed PCGF6 indeed interacted with MYC and MYC also interacted with PCGF6 in both human PSCs (endogenous interaction) and 293T cells with tagged PCGF6/MYC transfection (Fig. 5h and Supplementary Fig. 4g). Additionally, we purified both proteins with GST/His tag and performed the in vitro pull-down assay, demonstrating that MYC and PCGF6 physically interacted with each other (Fig. 5i). Moreover, the deletion of PCGF6 led to a significant reduction of MYC binding to SRE locus (Fig. 5j), while decreased expression of MYC did not affect the enrichment of PCGF6 on SRE or other PCGF6 and MYC-co-occupied target genes (*KLF1*, *OTX1*, *FOXO3*, *INPP5F* and *WDR36*) (Supplementary Fig. 4h–k). Importantly, knockdown of MYC indeed resulted in downregulation of SOX2 (Fig. 5k). These results indicate that PCGF6 recruits MYC to SRE and together activate SOX2 transcription.

By analyzing our PCGF6 and MYC ChIP data in human PSCs, we found there was a significant overlap between PCGF6-activated genes and MYC-binding genes (Supplementary Fig. 5a and Supplementary Data 2). Interestingly, deletion of PCGF6 significantly decreased the enrichment of MYC on these PCGF6-MYC shared genes (Supplementary Fig. 5b). Moreover, interrogation of additional chromatin marks or transcription factors indicated that a large majority of PCGF6-MYC shared binding sites overlapped with active chromatin H3K27ac, H3K4me3, P300, DNaes1, POL2 and CTCF peaks (Supplementary Fig. 5c). Collectively, these results demonstrated that PCGF6 mediates the recruitment of MYC to SRE locus, which in turn activates the transcription of SOX2 to promote neuroectoderm differentiation in human PSCs.

## Discussion

PcG proteins are critical epigenetic regulators of gene repression involved in various developmental processes[43–45]; however, little is known about how individual PcG proteins regulate gene expression, particularly during early human embryonic development. In this study, we have generated PCGF6-KO human PSCs using CRISPR/Cas9 technology and revealed the role of PCGF6 in human PSCs. Although these PCGF6-deficient human PSCs retain colony-formation capacity and alkaline phosphatase activity, and express high levels of pluripotent markers (*OCT4* and *NANOG*), they display aberrant expression of genes and partial differentiation compared with wild-type human PSCs. Furthermore, our data also demonstrate that the loss of PCGF6 in human PSCs leads to the promotion of mesendodermal differentiation and the impairment of neuroectoderm differentiation. Therefore, PCGF6 appears to be an important regulator of lineage-specification in human PSCs (Fig. 6).

As with all members of PCGF family, PCGF6 plays a role in transcriptional silencing of target genes via physically recruiting RING1B and then modifying histone in various biological processes[23,46,47]. Previous studies have shown that PCGF6 is highly expressed during mouse early embryonic development, and acts as an important regulator to maintain mouse ESC identity via regulating the expression of key pluripotency factors and germ cell-related genes[24,25]; however, our data indicated the maintenance of human PSCs was not altered by PCGF6 deficiency, and the expression level of key pluripotency factors did not differ between PCGF6-KO and wild-type human PSCs (Fig. 1). These observations suggest that the effect of PCGF6 on self-renewal can vary profoundly among cells of different species, or different pluripotent states. In addition, our data showed that the loss of PCGF6 in human PSCs leads to increased expression of the WNT superfamily members (Fig. 3), such as *FZD9* and *AXIN2*, and skewed differentiation into the mesendoderm lineage upon differentiation (Fig. 2e and Supplementary Fig. 2). ChIP-seq data revealed that the genomic loci of these WNT signaling pathway genes were co-occupied by PCGF6, E2F6, MAX and RING1B, which are components of PRC1.6 complex. Collectively, these observations demonstrate that PCGF6 acts as a repressor

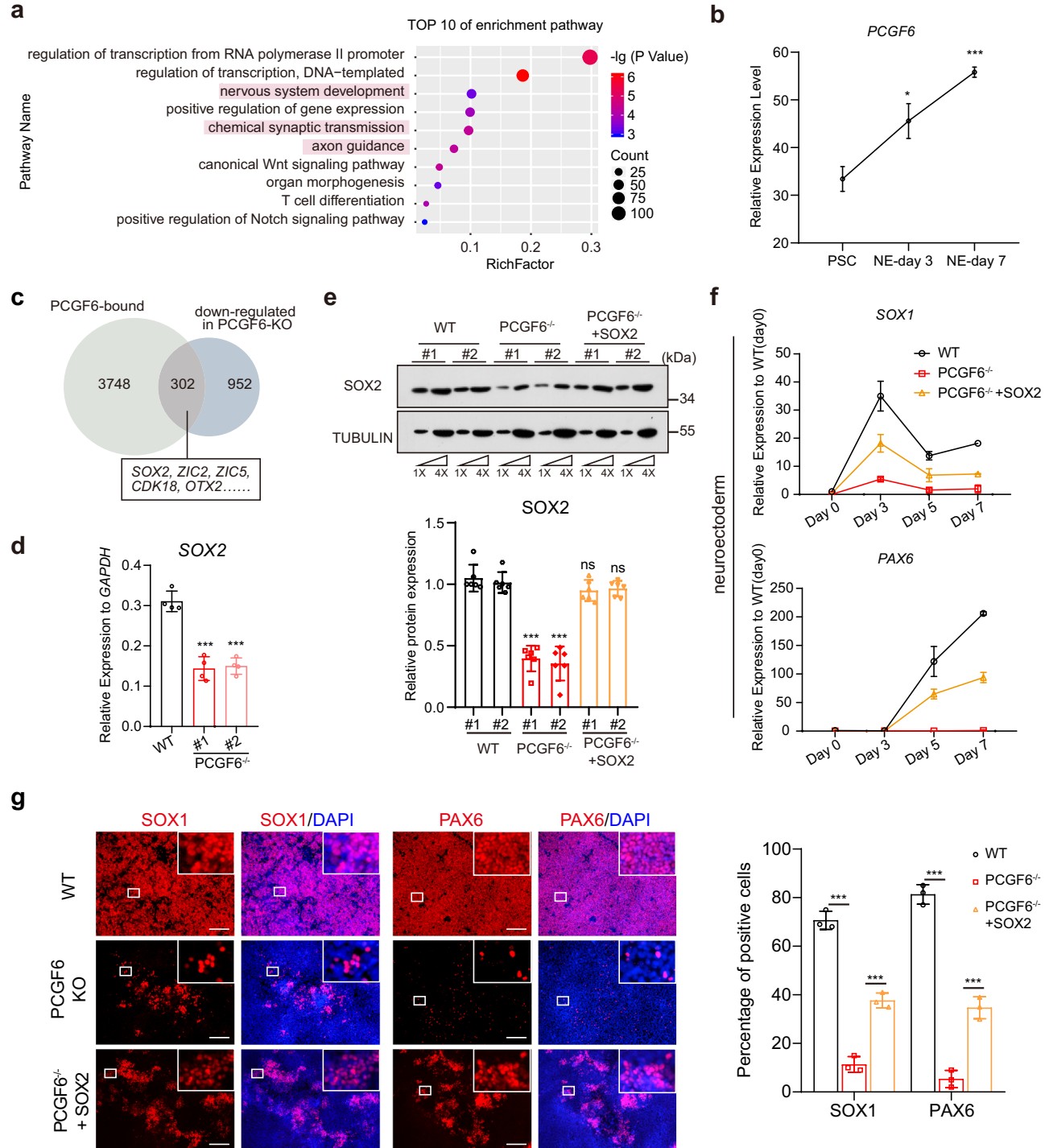

**Fig. 4 | PCGF6 is required for neuroectoderm differentiation from human PSCs through regulating SOX2. a** GO enrichment and KEGG pathway analysis of PCGF6-bound genes. **b** Expression analysis of PCGF6 during neural differentiation (n = 3). **c** Venn diagram shows the overlap of PCGF6-binding genes and downregulated genes in PCGF6-KO cells, revealing *SOX2* and its downstream genes as potential direct targets of PCGF6. **d** Relative expression levels of *SOX2* in the WT and PCGF6-KO human PSCs (n = 3). **e** Protein level of SOX2 in WT, PCGF6-KO and PCGF6−/− + SOX2 (PCGF6-KO/SOX2-overexpressing) human PSCs was determined and quantified; TUBULIN was used as a loading control (n = 3). **f** qRT-PCR analysis for expression of neural markers (*SOX1* and *PAX6*) in WT, PCGF6-KO and PCGF6−/− + SOX2 cells during neural differentiation. Results are shown relative to undifferentiated WT (n = 2). **g** Immunostaining of the neural progenitor markers (SOX1 and PAX6) in differentiated WT, PCGF6-KO and PCGF6−/− + SOX2 cells. Scale bars, 200 μm; the percentage of SOX1- or PAX6-positive cells was calculated (n = 3). Each point represents a biological replicate. Data are presented as the mean ± SD. Statistical significance was determined using the unpaired, two-tailed t-test in **b**, **d−g** (ns not significant, *p < 0.05, **p < 0.01, ***p < 0.001). Exact p values and statistical parameters are provided as a Source Data file.

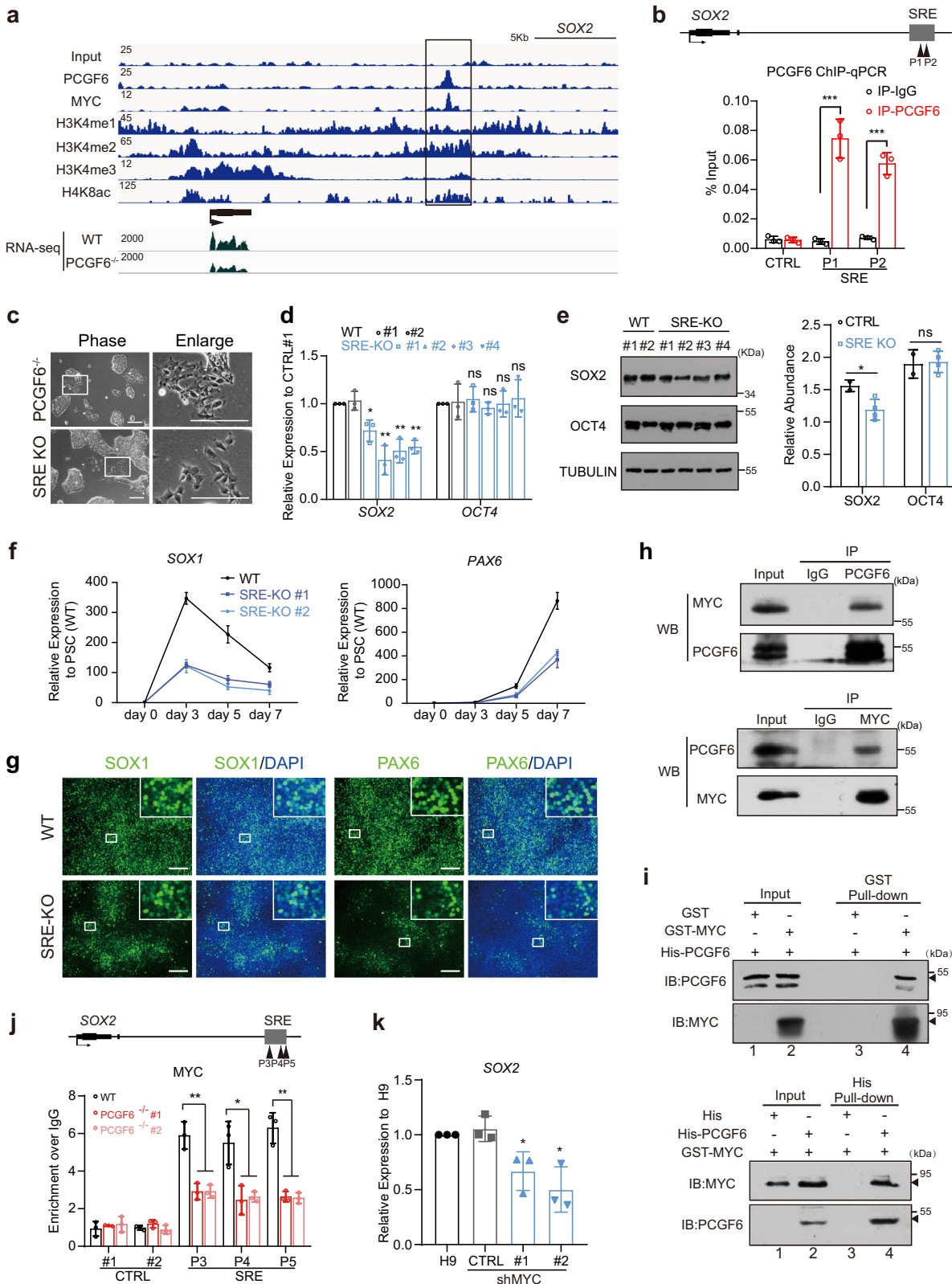

of genes that are involved in the WNT signaling pathway through PRC1.6-dependent inhibition. A great deal of evidence has proved that the activation of WNT signaling pathway is closely associated with mesendodermal specification during human PSCs differentiation[32,48,49]. Thus, based on our experimental data, we concluded that the function of PCGF6 in the repression of mesendodermal differentiation is mediated, at least in part, through the inhibition of WNT signaling pathway in human PSCs.

Recently, a few of reports had suggested that PcG proteins might also act as transcriptional activators during different development processes[50–54]. Here, we identified a bunch of down-regulated genes in PCGF6-KO human PSCs, including *SOX2* and its

**Fig. 5 | PCGF6 activates the expression of SOX2 via recruiting MYC. a** Bedgraph for PCGF6, MYC, H3K4me1, H3K4me2, H3K4me3 and H4K8ac bound at *SOX2* locus. The *x* axis corresponds to genomic locations with the scale indicated at the top of panel, and the *y* axis corresponds to ChIP-seq signal intensity. **b** ChIP-qPCR analysis for PCGF6 binding at SRE in human PSCs. IgG served as a negative control. ChIP enrichments are normalized to input (*n* = 3). Arrowheads represent the genomic position of qPCR primers. **c** Representative images showed the morphology of PCGF6-KO and SRE knockout (SRE-KO) human PSCs (*n* = 4). Scale bars, 200 μm. **d** Relative expression levels of *SOX2* and *OCT4* in the WT and SRE-KO human PSCs. The level of gene expression in the control is set as 1. #1–#4 represent four sub-clones of SRE-KO cell lines (*n* = 3). \**p* < 0.05 and \*\**p* < 0.01, determined by Student's *t* test. **e** Western blot analysis of SOX2 and OCT4 in control (*n* = 2) and SRE-KO (*n* = 4) human PSCs. TUBULIN served as a loading control. **f** qRT-PCR analysis for expression of neural markers (*SOX1* and *PAX6*) in the WT and SRE-KO cells during

neuroectoderm differentiation. Results are shown relative to undifferentiated WT (*n* = 3). **g** Immunostaining of the neural progenitor markers (SOX1 and PAX6) in the WT and SRE-KO cells after neuroectoderm differentiation (*n* = 3). Scale bars, 200 μm. **h** Immunoprecipitation data showed the interaction between endogenous PCGF6 and MYC in human PSCs (*n* = 3). **i** GST/His pull-down for purified GST-MYC protein and His-PCGF6 protein. Arrowheads indicate GST-fusion and His-fusion protein (*n* = 3). **j** ChIP-qPCR analysis for MYC-binding in WT and PCGF6-KO human PSCs at SRE. ChIP enrichments are normalized to IgG control (*n* = 3). Arrowheads represent the genomic position of qPCR primers. **k** qRT-PCR analysis for expression of *SOX2* in the WT and MYC-knockdown human PSCs. Results are shown relative to wild-type H9 (*n* = 3). Each point represents a biological replicate. Data are presented as the mean ± SD. Statistical significance was determined using the unpaired, two-tailed *t*-test in **b**, **d**–**f**, **j**, **k** (ns not significant, \**p* < 0.05, \*\**p* < 0.01, \*\*\**p* < 0.001). Exact *p* values and statistical parameters are provided as a Source Data file.

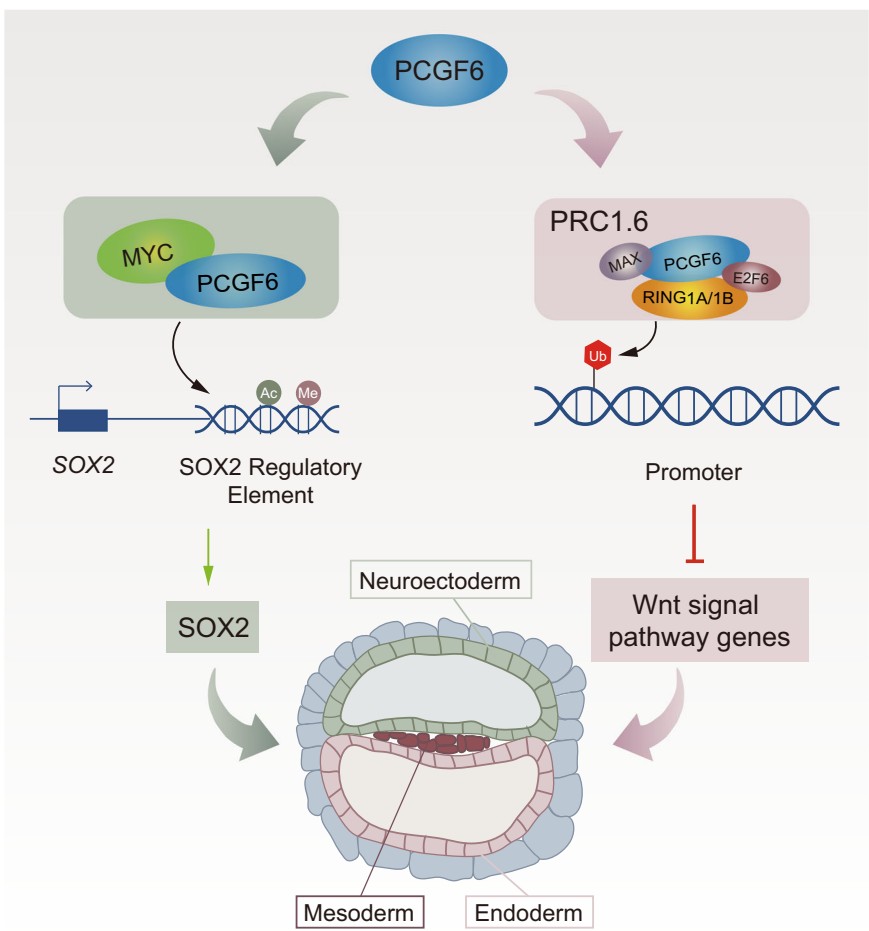

**Fig. 6 | Model for the role of PCGF6 in human PSCs.** PCGF6 plays dual functions in regulating human PSCs differentiation. PCGF6 not only recruits MYC to activate SOX2 expression that maintain early neuroectoderm differentiation, but also

resists skewed differentiation toward mesendoderm through the PRC1-dependent inhibition of WNT/β-catenin signaling pathways.

downstream genes especially. SOX2 is known to participate in neuroectoderm development, and the deficiency of SOX2 impairs the specification of the neural lineage in early embryonic development[40,41]. Thus, we investigated whether the aberrant phenotype of neuroectoderm differentiation is driven by lower expression of SOX2 after deleting PCGF6. Our data indicated that overexpression of SOX2 can rescue neuroectoderm differentiation defect in PCGF6-KO human PSCs, strongly indicating SOX2 as the functional target of PCGF6 (Fig. 4). Intriguingly, Pcgf6 has been reported that can replace Sox2 in reprogramming process[55], which further supports our conclusion that the regulatory relationship

between PCGF6 and SOX2. Moreover, we observed that PCGF6 physically binds to the downstream of SOX2 gene body, suggesting that it directly activates the expression of SOX2 in human PSCs. It is worth noting that the absence of the PCGF6 occupied region (SRE) led to a decline in the expression of SOX2 both at transcription and protein levels (Fig. 5), implying a potential new regulatory element of SOX2.

Intriguingly, PRC1 proteins have been recently reported to cooperate with transcription factors to activate gene transcription. For instance, PHC1 interacts with NANOG and maintains pluripotency through regulating chromatin architecture in human PSCs[56]. PCGF5

interacts with Tex10 and p300 and positively regulates expression of genes involved in mesoderm differentiation[53]. E2F6 triggers DNA methylation-dependent silencing of germline genes during development via indirect recruitment of DNMT3B[57]. Similarly, our data also revealed that PCGF6 interacted and occupied with MYC at the SRE locus, and activated the expression of SOX2 in human PSCs, which contribute to neuroectoderm development. Interestingly, there were a significant overlap between PCGF6-activated genes and MYC-bound genes. These PCGF6/MYC co-activated genes prefer to a more open chromatin conformation enriched with P300, POL2 and CTCF (Supplementary Fig. 5c–e). We speculate that in a given environment, MYC may act as a functional antagonist of PcG complexes, and competitively bind to element of PcG complexes to regulate gene expression. Very recently, EZH2 was reported to directly bind MYC via a cryptic transactivation domain and mediate gene activation[58]. Nevertheless, our finding supports that specific PcG proteins might cooperate with specific transcription factor and play distinct roles in the regulation of various biological process, yet not confined to its well-established repressor functions.

Traditionally, PcG proteins have been reported to be involved in either barriers or accelerators in developmental processes[4,59]. For instance, Kdm2b restrains early differentiation in mouse ESCs via repression of lineage-specific genes[60]; Mel18 functions as a facilitator for cardiac differentiation through both a context- and stage-specific manner[17]. Of particular interest, our results demonstrate PCGF6 appears to be an "on-off" switcher, which determines lineage specification along different trajectories. From a developmental point of view, PCGF6 acts as a barrier to mesendoderm differentiation, and acts as an accelerator to neuroectoderm differentiation during early embryonic development. Such epigenetic switchers have been previously described in very few studies. Human primed PSCs without PRC2 undergo spontaneous differentiation toward the mesendoderm germ layers but fail to neural ectoderm fate due to the derepression of BMP signal[61]. Similarly, EED has been studied as a direct repressive barrier for the secretory lineage commitment and an accelerator for progenitor cell proliferation in the intestinal epithelium[62]. Altogether, it is beginning to be noticed that PcG proteins are involved in both barriers and accelerators during diverse developmental processes.

In summary, this study highlights a critical lineage-specific function and mechanism for PCGF6 in balancing lineage specification during early human embryonic development. PCGF6 not only acts as a repressor to directly regulate the level of WNT signaling pathway genes in human PSCs, which in part impact mesendoderm differentiation, but also cooperates with MYC and positively regulates the expression of SOX2 that is required for neuroectoderm differentiation (Fig. 6). The present work provides new insights into the role of PcG family in the cell fate determination of human PSCs, which should facilitate the understanding of the diverse functions of epigenetic factors during cell fate determination and embryonic development.

## Methods

### Cell culture
Human induced pluripotent stem cell PGP1 and human ESC H9 (also WA09) were cultured on Matrigel-coated plates (Corning, Cat#354277) at 37 °C with 5% CO$_2$ in mTesR1 (STEMCELL Technologies, Cat#85850). These cells were passaged every 4 days by Accutase (STEMCELL Technologies, Cat#07922). The culture medium was changed every day. Human embryonic kidney 293T (293T) cells were cultured on tissue-cultured plates with DMEM containing 10% FBS (Gibco, Cat#10100147) and 1x penicillin/streptomycin (Gibco, Cat#10378016) at 37 °C with 5% CO$_2$ in cell incubator. These cells were passaged by 0.05% trypsin (Gibco, Cat#25300120) at 90–100% cell confluence.

### Generation of knockout cell lines
The pair gRNAs were designed on the website (http://crispr.mit.edu/) to knockout PCGF6 and SRE (SOX2 regulatory element). We used the pX459 that contained Cas9 and gRNA sequence. Briefly, one million cells were electroporated with 5 ug pX459 containing sgRNA1 and sgRNA2 respectively. Then the cells were seeded on Matrigel-coated 6-well plate in mTesR1 medium with 10 μM Y-27632 (Selleck, Cat#S1049). After 24-h culturing, the puromycin (1 μg/ml) (Santa Cruz Biotechnology, Cat#58582) was added for another 48 h and then single cells were replated into a Matrigel-coated 96-well plates by FACSAria III. The survived colonies were expanded and genotyped by Sanger sequencing. Finally, we generated PCGF6 homozygous knockout cell lines in PGP1 and SRE homozygous knockout in H9. The sequences of two gRNAs targeting PCGF6 are GTAGGCGCTGCCAAAACCGA and CCGCTTCGAGAGGCCGCTTCG. The sequences of two pairs of gRNAs deleting SRE are TAAAAGTGCCATTTTTTCC and TCGCGG TAAGAGCAGAAAAT, CGCAGGAAGGAAAACCATAA and ATAGCCGG TTAATTCCCCCA, respectively.

### MYC knockdown
The two lentiviral vectors containing the shMYC sequence are the gifts from Prof. Guoliang Qing Lab at Wuhan University (MYC-sh#1: CAGTTGAAACACAAACTTGAA; MYC-sh#2: CCTGAGACAGATCAGCAA CAA). The lentivirus was packaged in HEK293T cells and concentrated. After transfection into human ESCs, puromycin was used to select stable MYC knockdown cells. The survival cells were expanded and confirmed by western blot and RT-qPCR.

### Overexpressing SOX2 and PCGF6 in PCGF6-KO human PSCs
Lentiviral vectors (with GFP report) with the target gene (SOX2, or PCGF6) was packaged in HEK293T cells and infected the PCGF6-KO PSCs. After 48 h, the GFP+ cells were sorted in FACSAria III and expanded.

### Alkaline phosphatase (AP) staining
Human PSCs were cultured on Matrigel-coated 24-well plates for 2 days, and AP staining was performed according to manufacturer's recommendations (Beyotime, Cat#C3206). Briefly, cultured cells were fixed with 4% paraformaldehyde at room temperature for 20 min. Then the cells were washed for three times with DPBS and incubated in BCIP/NBT staining mix in dark for 3 h at room temperature. After washing with DPBS for 1–2 times to terminate the staining reaction, the cells were counterstained with neutral red staining solution for 30 min. Colonies were visualized and photographed using microscopy.

### Western blot
Whole cell extracts were obtained with RIPA buffer (Beyotime, Cat#P0013C) supplemented with protease inhibitor cocktail (Roche, Cat#4693116001). The lysates concentration was quantified by the BCA protein assay kit (Thermo Fisher Science, Cat#A53225). About 30 μg lysate proteins were loaded onto 10% SDS-PAGE, and transferred to a nitrocellulose membrane (Millipore, Cat#Z746010). Then the membrane was washed with 5% (w/v) skimmed milk in 0.1% Tween-20 TBST and incubated with the indicated antibodies overnight at 4 °C. After washing with 0.1% tween-20 TBST for 30 min, the membranes were incubated with HRP-conjugated second antibodies for 1 h at room temperature, followed by Immobilon Western Chemiluminescent HRP Substrate (Millipore, Cat#WBKLS0500) treatment and visualized. The used antibodies include: anti-PCGF6 (1:1000, Proteintech, Cat#24102-1-AP), anti-GAPDH (1:5000, Proteintech, Cat#60004-l-lg), anti-α-Tubulin (1:5000, Proteintech, Cat#11224-1-AP), anti-OCT4 (1:1000, Santa Cruz Biotechnology, Cat#sc-5279), anti-SOX2 (1:100, Santa Cruz Biotechnology, Cat#sc-20088), anti-NANOG (1:1000, Santa Cruz Biotechnology, Cat#sc-

33759), anti-MYC (1:1000, Cell Signaling Technology, Cat#13987S), anti-EHMT2 (1:2000, Proteintech, Cat#66689-1-Ig), anti-L3MBTL2 (1:2000, Proteintech, Cat#39570), anti-MAX (1:2000, Proteintech, Cat#10426-1-AP), anti-RYBP (1:2000, Proteintech, Cat#11365-1-AP), anti-E2F6 (1:2000, ABclonal, Cat#A2718), and anti-HDAC1 (1:2000, Proteintech, Cat#10197-1-AP).

## Immunofluorescence assay

Cultured cells were fixed with 4% paraformaldehyde at room temperature for 15 min and blocked with blocking solution (DPBS with 10% (v/v) donkey serum and 0.3% Triton-100). Then the cells were incubated with primary antibodies overnight at 4 °C, followed with washing and staining with secondary antibodies for 2 h in dark at room temperature, and then counterstained with 5 µg/ml 4′,6-Diamidino-2-phenylindole dihydrochloride (DAPI, Sigma, Cat#10236276001) for 10 min. Cells were visualized and imaged using fluorescence microscopy (Olympus). The antibodies for immunofluorescence assay used in present work include: anti-PAX6 (1:200, Biolegend, Cat#l901301), SOX1 (1:200, Cell Signaling Technology, Cat#4195S), anti-OCT4 (1:200, Santa Cruz biotechnology, Cat#sc-5279), anti-NANOG (1:200, Cell Signaling Technology, Cat#4903), anti-PDX1 1:200, abcam, Cat#ab47267), anti-T/Brachyury (1:200, Cell Signaling Technology, Cat#81694S), anti-TUJ1 (1:200, SIGMA, Cat#T3952).

## Endoderm differentiation

The DE differentiation was performed according a previously described protocol[28]. Briefly, $5 \times 10^4$ cells were seeded onto Matrigel-coated 24-well plates in mTesR1. On the next day, these cells were cultured in DE induction medium (DMEM-F12 (Gibco, Cat#C11330500BT), 0.2% BSA (YEASEN, Cat#B57370) added with Activin A (100 ng/ml, Pepro-Tech, Cat#120-14P). The cells were collected for immunofluorescence assay, qPCR assay and flow cytometry analysis at day 4.

## Pancreatic progenitor differentiation

Based on our previous protocol[29], the differentiated definitive endoderm cells were cultured in MCDB131 (Sigma), supplemented with 1.5 g/l sodium bicarbonate, 10 mM glucose (Invitrogen), 0.5% BSA, 0.25 mM ascorbic acid, 1× ITS-X, 1× GlutaMAX, and 1% penicillin-streptomycin. In total, 50 ng/ml KGF, 0.5 µM SANT1, 100 nM TTNPB, 500 nM PDBU, 200 nM LDN193189, and 2 µM IWR-1 were added to this basal medium for 8 days.

## Cardiomyocyte differentiation

The cardiomyocyte differentiation was performed according a previously described protocol[30]. Briefly, $2.5 \times 10^4$ cells were seeded onto Matrigel-coated 24-well plates in mTesR1. After 2 days, cells were treated with 2.5 µM CHIR99021 (Selleck, Cat#S2924) in RPMI 1640 (Thermo Fisher, Cat#C11875500BT) supplemented with B27-without insulin. After 48 h cells were treated with 2.5 µM IWP2 diluted in RPMI-B27 without insulin and incubated for 5 days. At day 7 the medium was changed to RPMI-B27 with insulin without any extra supplement and medium was changed every 2 days thereafter.

## Neuroectoderm differentiation

The neuroectoderm differentiation protocol was performed according to a previous report[42] with modifications. Human PSCs were dissociated with Accutase to obtain single cells and then seeded onto Matrigel-coated 24-well plates with the density of $4 \times 10^4$ cells per well. After 1- or 2-days' culture in mTesR1, cells were treated with neural induction medium DMEM/F12, 0.5× N2 (BasalMedia, Cat#S430J4)), 0.5×B27 (BasalMedia, Cat#S441J7), 1% Gluta-max, 1% NEAA, 0.1% β-mercaptoethanol) plus 10 µM SB431542 (Selleck, Cat#S1067) and 0.5 µM LDN193189 (Selleck, Cat#7507) to initiate neuroectoderm differentiation. After 5 days of neural induction, SB431542 was withdrawn

for additional 2 days. Fresh culture medium was changed every day. The cells were collected for immunofluorescence assay at day 3 and 7, and for qPCR assay at day 0, 3, 5 and 7.

For neuronal differentiation, the generated neuroectoderm progenitors were further treated with BDNF (MCE, #HY-P7116A), ascorbic acid (Sigma, #A4544), and FGF2 (PeproTech, #100-18B) for another 2 weeks.

## Embryoid body (EB) formation

Human PSCs were treated with Accutase to obtain single cells, and then counted and resuspended with the density of 100 cells/µl in mTesR1 with 10 µM Y-27632. Then the single cell drops (20 µl) were hanging on the lid of Petri Dishes. One or two days later, the aggregated EBs were collected into 6-well low attachment plates. Then the culture medium was changed to EB medium (DMEM, 10% FBS, 1% penicillin-streptomycin). The EB medium was changed every 2 days. After 9 days' spontaneous differentiation, the differentiated EBs were collected for qPCR assay.

## Flow cytometry analysis

For flow cytometry analysis, TrypLE Express (Gibco, Cat#12604013) was used to detach cultured cells. Then the cells were washed twice with DPBS containing 2% FBS and incubated with APC-conjugated mouse-anti-human CD184 (BD, Cat#555976) or APC-conjugated mouse isotype IgG (BioLegend, Cat# 400220) in dark for 30 min at 4 °C. After washing twice, the cells were resuspended by DPBS and analyzed on the CytoFLEX flow cytometer (Beckman).

## MTT (3-(4,5-dimethylthiazol-2-yl)−2,5-diphenyltetrazolium bromide) assay

Single cells were seeded with the density of 2500 per well in a 96-well plate coated with Matrigel. After culturing for 48, 72 and 96 h, 10% MTT was added into culture medium for another 4 h incubation at 37 °C incubator. Then the MTT-medium was removed and 200 µl DMSO (dimethyl sulfoxide) was added into each well for another 10 min. Lastly, the plate was shaking on an orbital shaker and the absorbance value was measured at 490 nm with MD SpectraMax i3x.

## RNA preparation and qRT-PCR

Cells were rinsed in DPBS and total RNA was extracted with HiPure Total RNA Mini Kit (Magen, Cat#R4111-03). One µg RNA were used to reverse to cDNA with the ABScript II RT Master Mix (ABclonal, Cat#RK20402) according to the manufacture's protocol. Then the cDNAs were used for real-time PCR with 2x SYBR Green qPCR Master Mix (Biomake, Cat#B21203) and performed on a C1000 Touch Thermal Cycler machine (Bio-Rad). The relative gene expression was normalized to *GAPDH* based on the delta Ct method. All Real-Time qPCR experiments were carried out at least three replicates. Comparison between samples was performed using Student's *t* test. The primers used in the qRT-PCR assays are listed in Supplementary Tables 1 and 2.

## Co-immunoprecipitation

One ml of cell lysates was mixed with 1x cOmplete Protease Inhibitor and sonicated with ultrasonic homogenizer (Scientz bioscience) for 5 min of 2 s/on and 4 s/rest. After clarified by centrifugation, the supernatant was incubated with 5 ug antibodies at 4 °C for rotating overnight, and then with Protein A/G Dynabeads at 4 °C for additional 4 h. The immunoprecipitates were then washed once with RIPA and twice with wash buffer (20 mM Tris-HCl, 500 mM NaCl, 2 mM EDTA, 1% Triton X-100, 0.1% SDS, pH 8.0), and resuspended in 40 µl of 1× SDS loading buffer, followed by western blot. Rabbit IgG antibody was used for controls. The used antibodies include: anti-PCGF6 (Proteintech, Cat#24102-1-AP), anti-MYC (Cell Signaling Technology, Cat#13987S), anti-RING1B (Cell Signaling Technology, Cat#5694).

## RNA-seq and data analysis

RNA was quantified by a DNA/Protein Analyzer (QuaWell) and sent to Geekgene (Beijing, China) for RNA-seq library construction and sequencing. For data analysis, reads were qualitied by FastQC (v0.11.9) and the adapter was trimmed by Trim_galore (v0.6.2). The clean reads were aligned to the human reference genome GRCh38 using HISAT2 (v2.1.0), and gene expression counts were determined by featureCounts (v1.6.4). To determine differential expression genes, DESeq2 (v1.20.0) was used to define significant differences genes by setting adjust $p$ value <0.05 and abs|$\log_2$(fold-change)| > 0.6. After obtaining the significant differences genes, gene ontology analysis was performed on DAVID (https://david.ncifcrf.gov/) and the heatmap was made by heatmap (v1.0.12).

## ChIP-seq and data analysis

Approximately $1 \times 10^7$ cells for each sample were crosslinked with 1% formaldehyde for 10 min at room temperature and then stopped by incubating with 0.125 M glycine for 5 min. After washed twice with cold DPBS, cells were collected by cell scraper and centrifuging at 4 °C, and then, the cell pellets were re-suspended in Cell Lysis Buffer (10 mM Tris-HCl pH 8.0, 140 mM NaCl, 0.2% NP-40) plus 1x cOmplete Protease Inhibitor. The isolated nuclei were collected and sonicated in nuclear lysis buffer (50 mM Tris-HCl (pH 8.0), 10 mM EDTA (pH 8.0), 1% SDS) with 1x cOmplete Protease Inhibitor in Diagenode Bioruptor Pico for ten cycles with 20 s/on and 30 s/off to obtain 200–500 bp chromatin fragments. Then, the soluble sonicated fragments were incubated with 5 μg antibody and Protein A/G Dynabeads with rotating overnight at 4 °C. On the following day, the beads were washed, and the DNA fragments were eluted with the elution buffer and reverse-crosslinked at 67 °C for 5 h. Finally, the immunoprecipitated DNA were purified with phenol-chloroform and sent to Geekgene (Beijing, China) for ChIP-seq library construction and sequencing. The used antibodies include: anti-PCGF6 (1:200, Proteintech, Cat#24102-1-AP), anti-H2AK119Ub (1:200, Cell Signaling Technology, Cat#8240), anti-MYC (1:1000, Cell Signaling Technology, Cat#13987S).

For ChIP-seq data analysis, reads were qualified by FastQC (v0.11.9) and the adapter was trimmed by Trim galore (v0.6.2). Then these clean reads were aligned to the human reference genome GRCh38 using Bowtie2 (v2.3.4.2) with end-to-end. The uniquely mapped reads were sorted by Samtools (v1.9) and then performed peak calling by MACS2 (v2.1.2) with the callpeak module (the parameter–call-summits–extsize 150 -p 0.05). The most significantly enriched peaks were selected with a threshold by corrected Fold-change >3. The bam files were converted to the bigWig signal files by DeepTools (v3.3.0) with the bamCoverage module and then visualized using the DeepTools module of computeMatrix, plotHeatmap and plotProfile. Homer was used to annotated nearby genes from the peaks obtained from MACS2 and to find motif that enrich in the binding sequence. Gene ontology analysis was performed on DAVID (https://david.ncifcrf.gov).

## Luciferase reporter assay

The luciferase reporter assay was performed as previously described[28]. PCGF6 was cloned into the pL-GFP vector. Next, the HEK-293T cells were cultured into 96-well plates and co-transfected with PCGF6-pL-GFP, the pGL4.17-basic luciferase reporter plasmid and the pRL-TK Renilla luciferase reporter plasmid, and the pL-GFP vector was used as the control. Transfection was performed with Sage Lipoplus DNA Transfection reagent (Sage, Cat#Q03004). After 6-h transfection and 24-h culture, the cells were collected, and luciferase activity was measured with the Luciferase Assay Kit (Promega, Cat#E1910). Each experiment was performed independently three times.

## GST/His pull-down

To detect the direct interaction between PCGF6 and MYC proteins, the GST/His pull-down assay was performed as previously described with some modifications[63]. The prokaryotic expression vectors are gifts from Prof. Qiang Chen Lab at Wuhan University. In brief, the recombinant GST-MYC and His-PCGF6 proteins were purified from E. coli BL21 (DE3) cells by standard protocols. After incubated with 20 μl Glutathione Sepharose 4B (Abclonal, Cat#AS044) for 2 h at 4 °C, the GST (20 μg) or GST-MYC proteins (20 μg) were further incubated with purified His-PCGF6 protein (20 μg) for at least 4 h. With low-speed centrifugation, the beads were washed five times in washing buffer. Then proteins were eluted from the glutathione-agarose and resolved by 12% SDS-PAGE. Western blot was performed to detect protein with the indicated antibodies.

## Cell cycle and cell colony analysis

The cell cycle analysis was performed according to a previous work[64]. In brief, one million single cells were fixed in 75% ethanol at 4 °C overnight. After washing the cells once with DPBS, 500 μl DPBS with 100 μg/ml RNase A was added to resuspend the cell for incubation at 37 °C for 30 min. Then the cells were incubated with 50 μg/ml propidium iodide (PI) at room temperature for another 30 min in dark. Finally, a FACSCelesta flow cytometer (BD Biosciences, USA) was used to analyze these cells. These cells were divided into each cell cycle phase (subG1, G1, S and G2/M) based on the PI intensity in the FlowJo (v10.4.0) and calculated the proportion. The cell colony size was calculated in the ImageJ and the significance level was calculated by two-tailed unpaired $t$-test in GraphPad Prism 8.

## Statistics and reproducibility

The experiments were repeated independently as described in the figure legends. Data are given as mean ± SD. The significance level was calculated by Student's $t$ test in GraphPad Prism 8 and $p$ values are shown with *$p < 0.05$, **$p < 0.01$, ***$p < 0.001$. Uncropped and unprocessed scans of blots are provided in Supplementary Information.

## Reporting summary

Further information on research design is available in the Nature Research Reporting Summary linked to this article.

# Data availability

The RNA-seq and ChIP-seq data generated in this study have been deposited in the Gene Expression Omnibus (GEO) database under accession code GSE173690. The processed RNA-seq and ChIP-seq data are available at GEO database. The remaining data are available within the article (and Supplementary information). Uncropped Western blots and Source data are provided with this paper.

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

## Acknowledgements

We would like to thank Prof. Yi Zhang (Harvard Medical School) for insightful discussion and editing help, Prof. Donghui Zhang (Hubei University) for help in figure preparation, Dr. Liyuan Wang and Prof. Guoliang Qing (Wuhan University) for technical help, and Chenchao Yan, Ran Liu, Mao Li, Jie Yang, Yajing Meng, Zhinang Yin and other laboratory members for technical help and helpful discussion. We thank the core facility of the Medical Research Institute at Wuhan University for the technical support. This work was supported by the National Key Research and Development Program of China (No. 2016YFA0503100), the Science and Technology Department of Hubei Province Key Project (2020CFA017, 2021CFA049), Health Commission of Hubei Province scientific research project (WJ2021Q029), and the Fundamental Research Funds for the Central Universities in China (2042021kf0207).

## Author contributions

Conceptualization: W.Jiang and X.L.; Methodology: W.Jin, C.L., and X.L.; Software: S.D. and T.Z.; Formal analysis: X.L.; Resources: Y.Y., K.L. and J.C.; Writing—original draft: X.L.; Writing—review and editing: W.Jiang, H.W., X.L. and S.D.; Funding acquisition: W.Jiang; Supervision: W.Jiang.

## Competing interests

The authors declare no competing interests.
