## [Peer Review File · Nature Communications]

PCGF6 controls neuroectoderm specification of human pluripotent stem cells by activating SOX2 expressionREVIEWER COMMENTS

Reviewer #1 (Remarks to the Author):

In this manuscript, Lan et al. show that the PRC1 subunit PCGF6 plays a key role in dictating cell fate determination on hESC. Specifically, they demonstrated that PCGF6 prevents pluripotent cells to differentiate towards the mesoendoderm lineage. PCGF6 depletion biases differentiation towards mesoendoderm differentiation and prevents neuroectoderm differentiation. Mechanistically, they show that while PCGF6 exerts its canonical function within the PRC1 complex it can also mediate transcriptional activation of a set of neuroectodermal genes including SOX2. Analysis of signaling pathways deregulated upon PCGF6 depletion revealed a role of PCGF6 in regulating the WNT pathway which can explain the differentiation bias towards the mesoendoderm lineage. Finally, they demonstrated that PCGF6 interacts with MYC to regulate SOX2 expression, proving a mechanism by which PCGF6 regulates cell fate decisions.

This manuscript is interesting as it identifies new evidence supporting the role of PCGF6 in cell fate decisions and nicely complements with published work on the role of PCGF paralogs in stem cell differentiation (i.e., PCGF2 in mesoderm differentiation, PCGF4 in neural differentiation). Unfortunately, the novelty of this study is partially compromised by another study from Koseki's lab.

Overall, the experiments are well controlled and the conclusions are supported by the data (except the role of PCGF6 in self-renewal). I think that the paper could be improved if they dig more on the MYC-PCGF6 axis, differentiation assays and analysis of self-renewal. The current state of this manuscript is too preliminary to be published in Nature Communications.

Major points:

1. I do not totally agree with the conclusions of the authors regarding the effect of PCGF6 depletion in self-renewal. Although it seems clear that pluripotency factors are not downregulated upon PCGF6 loss, AP stainings in figure 1 suggest that PCGF6 null cells have partially differentiated colonies. Moreover, later in the manuscript (figure 5c) they show spontaneous differentiation of PCGF6 KO cells and SOX2 SRE KO lines. Additionally, Koseki and colleagues recently showed (ref 23) a role for PCGF6 in maintaining mouse ESCs in an undifferentiated state. The authors need to perform additional analysis to clarify the role of PCGF6 in maintaining self-renewal and preventing spontaneous differentiation. For instance, among other experiments, they could determine the size of the colonies in WT and PCGF6 KO cells and the cell cycle profile of WT and KO cells.

2. Unfortunately, the connection between PCGF6 and MYC is not entirely novel: 1) PCGF6 as part of PRC1.6 which interacts with the two bona-fide MYC interacting proteins MGA and MAX, and 2), Koseki and colleagues reported co-localization of PCGF6 and MYC in mouse ESCs. Therefore, additional experiments need to be performed to increase the novelty of the MYC-PCGF6 axis:

- a. Does MYC directly interact with PCGF6?
- b. Is the interaction mediated by nucleic acids?
- c. Is MYC required for PCGF6 recruitment genome-wide and to the SOX2 SRE?
- d. Is PCGF6 the only PRC1.6 subunit recruited to the SOX2 SRE?
- e. Is PCGF6 recruited to other active enhancers in pluripotent cells occupied by MYC?
- f. Is the recruitment of PCGF6 to active regulatory regions specific to MYC-occupied sites or is PCGF6 also found in enhancers occupied by other pluripotency factors?

3. Regarding the role of PCGF6 in cell fate decisions, the authors need to perform additional assays. For instance, it is not clear if PCGF6 is important for the generation of NPC (neuroprecursors) or differentiated neurons. The authors should test both hypotheses. In line with this, mesoendoderm differentiation is still a very premature differentiation event. Is PCGF6 important for repressing cardiac, hematopoietic differentiation or endoderm cell differentiation? They should test 2 or 3 differentiation assays to delineate more precisely the role of PCGF6 in cell fate decisions.

Other points:

1. Please show the stability of PRC1.6 upon PCGF6 depletion.
2. Figure 5e is not convincing as the SOX2 levels don't seem to decrease in the SRE KO cells
3. It would be important to show rescue of the neuroectoderm differentiation by reintroducing PCGF6.

Reviewer #2 (Remarks to the Author):

In this manuscript entitled "Polycomb-group ring finger 6 controls neuroectoderm specification of human pluripotent stem cells by activating SOX2 expression", Xianchun Lan et al. aims to demonstrate that the Polycomb group ring finger 6 protein (PCGF6) controls neural-ectoderm specification of human pluripotent stem cells (PSCs) by activating SOX2 gene.

The authors tested whether PCGF6 modulates pluripotency and lineage specification and contributes to early development in human cells. They tried to define the role of PCGF6 in the regulation of gene expression during early human embryonic development by generating a PCGF6 KO in human PSCs. PCGF6 KOs maintained high levels of pluripotent markers, the ability of colony formation as well as alkaline phosphatase activity but displayed aberrant neuroectoderm differentiation during differentiation suggesting an important role as regulator of lineage specification in PSCs. By performing RNA-seq analysis in KO PSC they found 970 genes upregulated that included several mesodermal and ectodermal genes. Among the downregulated genes they found an enrichment in genes involved in nervous system development, including SOX2 which was identified as a direct target of PCGF6.

Based on these data, the authors hypothesized that PCGF6 can function as lineage switcher between mesoderm and neuroectoderm in human PSCs by both suppressing and activating mechanisms. In particular, they found that PCGF6 and MYC directly interact and co-occupy a distal regulatory element of SOX2 to activate SOX2 expression, which likely accounts for the defects in neuroectoderm differentiation.

To investigate whether the binding of PCGF6 to SOX2 was the mechanism through PCGF6 may cause destruction of the neuroectodermal differentiation, they rescued PCGF6 KO neuroectoderm defects with SOX2 ectopic expression.

The manuscript is clear and technically well organized. However, although the role of PCGF6 and PRC1.6 in regulating the cell fate decision remains interesting, the experiments fell short in supporting the proposed mechanism and its novelty in sustaining the PCGF6 role in balancing the cell fate decision.

It is well established that PCGF6 is expressed at high levels in mouse embryonic stem cells (ES), where it is required for ESC identity. The mechanism involved still remains controversial. Two reports already suggested a repressive function of Pcgf6 on mesodermal-specific and on endodermal lineage genes, while Yang et al. suggested an PRC1.6-independent direct activator function of Pcgf6 on core ESC regulators such as Oct4, Sox2 and Nanog (PMID: 25187489 PMID: 27247273). These data revealed novel roles for Pcgf6 in directly regulating Oct4, Nanog, Sox2, and Lin28 expression to maintain ESC identity. PMID: 28304275

It is also well known that general lack of Polycomb activity in differentiating EBs results in neuroectodermal suppression and lack of pluripotency genes expression. Thus, phenotypically, the authors are presenting a classical Polycomb phenotype reproduced by several labs and papers in the past decades. The reduced expression of SOX2 could simply be an indirect consequence of a delayed/compromised differentiation. The overlap of both up or down regulated genes is poor and

seems rather stochastic. Furthermore, the re-expression of SOX2 may simply force the system to differentiate more efficiently, resulting in a partial rescue. A similar effect may also occur upon lack of other critical Polycomb regulators. Based on the model, one arm of PCGF6 function should be activatory together with MYC. First It's unclear what role should PCGF6 have in a complex with MYC. Such interaction was never reported before although PCGF6 has been extensively purified with proteomics approaches by several labs also in cells of neuronal lineages. This imply also that other Polycomb activities, PCGF6 co-factors as well as RING1A/B enzymatic activity should have no role in this phenotype. It is also unclear how MYC could interact with PCGF6. The model implies that this does not involve and depends on other Polycomb components , however this was not tested. Furthermore, PCGF6 forms a complex with the MGA/MAX dimer, which has affinity for E-Boxes and should be in competition with MYC DNA binding. The PCGF6 peak on SRE is not very convincing compared to snapshots on other loci and, considering that this is a bulk population of a dynamic heterogenous system, this result does not demonstrate a co-occupancy but rather represents the chromatin state of distinct cell populations suggesting a competitive behavior.

It is also important to highlight that it has been reported how loss of *Pcgf6* in mice results in partially penetrant embryonic lethality, clearly suggesting the critical roles for PRC1.6 subunits in the early cell fate decisions (PMID: 32482889). How precisely this occurs has to be elucidated but no links with the proposed in vitro phenotype are available. Overall, although the correlative gene ontology analyses goes in the direction of the proposed model, the manuscript remains incremental its first part and lacks several mechanistic information to support the proposed model.

Reviewer #3 (Remarks to the Author):

In this manuscript, Lan et al. investigated the function of PCGF6 in human PSCs maintenance and differentiation. The results showed that genetic deletion of PCGF6 did not affect the PSC self-renewal but favored the differentiation of PSCs towards the mesendodermal lineage. Genome-wide transcriptome and epigenome analysis showed that PCGF6, RING1B, RYBP bound to the promoters of Wnt signaling target genes and repressed their expression. In contrast, PCGF6 activated the *Sox2* gene expression by binding to its enhancer regions with MYC. Deletion of *Sox2*-regulatory element phenocopied the impaired neuroectoderm differentiation of PCGF6-deleted PSCs. The authors proposed that PCGF6 can function as lineage switcher between mesendoderm and neuroectoderm through either suppression or activation mechanisms.

Overall, the results in this MS are well presented and convincing. However, several issues need to be addressed:

Major points:

1. Different from mouse ESCs in which the PRC1.6-repressed targets are enriched with meiosis and germ cell-related genes, deletion of PCGF6 in human PSCs de-repressed much broader genes involving mesendodermal lineage development (Fig. 2A). It is unclear whether these genes are specifically repressed by PRC1.6 or by overlapped effects from other nPRC1 variants because (1) the ChIP-seq results in Figure 3e did not show that genome-wide PRC1.6-specific components such as E2F6 and Max/MGA co-occupied with PCGF6 and the data was only shown at two gene loci (Fig. 3g); (2) RING1B and RYBP are not PRC1.6-specific components; (3) in the PCGF6-KO cells, the uH2A signals was reduced but not completely lost, suggesting other PRC1-complexes still bound to chromatin and mediated uH2A. The authors should perform the ChIP-seq analysis using PRC1.6-specific components such as E2F6 and MAX/MGA in both wild-type and PCGF6-knockout cells to define the genes that are specifically regulated by the PCGF6-containing PRC1.6 complex before drawing the conclusion that PRC1.6 represses the mesoendodermal lineage-specific genes in human PSCs.
2. The finding that PCGF6 and MYC co-occupied at the *Sox2* enhancer was novel, and the ChIP-seq

qPCR results suggested that PCGF6 recruited MYC to the Sox2 enhancer. It was unclear whether this was a unique mechanism at the Sox2 locus. Would it be nice if the authors could do some analysis to examine whether the mechanisms found at the Sox2 locus are also utilized to regulate the 237 downregulated genes in the PCGF6-KO cells (Fig. 5h).

3. It is unclear what the mechanisms determine the recruitment of PRC1.6 or PCGF6-MYC at different genomic loci shown in the model (Fig.6). Would it be nice if the authors include some discussion in the discussion section?

Minor points:

1. Fig. 3e: RING1B signal is very low. The plots should also include the input control.
2. All photos missed the scale bars.

Response to reviewers' comments

We thank the reviewers for their insightful comments and constructive suggestions. To address the reviewers' concerns, we have performed additional experiments and data analyses, clarifying the statements in the previous version. The new data includes: 1) evidence for the direct interaction between PCGF6 and MYC proteins; 2) ChIP data analyses of PCGF6 and MYC co-occupancy on the active regulatory regions; 3) PCGF6 re-expression in PCGF6-knockout cells to confirm the phenotype; 4) multi-lineage differentiation assays demonstrating the important role of PCGF6 in cell fate decisions (favoring in neural lineage and repressing mesoderm and endoderm lineage); 5) the effect of PCGF6 knockout on PRC1.6 integrity and the enrichment of MAX and E2F6 on the genome. With these new data and editorial revision, we believe that our manuscript is now ready for publication in Nature Communications. Our point-by-point responses to reviewers' concern are listed below:

Reviewer #1 (*Remarks to the Author*):

In this manuscript, Lan et al. show that the PRC1 subunit PCGF6 plays a key role in dictating cell fate determination on hESC. Specifically, they demonstrated that PCGF6 prevents pluripotent cells to differentiate towards the mesoendoderm lineage. PCGF6 depletion biases differentiation towards mesoendoderm differentiation and prevents neuroectoderm differentiation. Mechanistically, they show that while PCGF6 exerts its canonical function within the PRC1 complex it can also mediate transcriptional activation of a set of neuroectodermal genes including SOX2. Analysis of signaling pathways deregulated upon PCGF6 depletion revealed a role of PCGF6 in regulating the WNT pathway which can explain the differentiation bias towards the mesoendoderm lineage. Finally, they demonstrated that PCGF6 interacts with MYC to regulate SOX2 expression, proving a mechanism by which PCGF6 regulates cell fate decisions.

This manuscript is interesting as identifies new evidence supporting the role of PCGF6 in cell fate decisions and nicely complements with published work on the role of PCGF

paralogs in stem cell differentiation (i.e., PCGF2 in mesoderm differentiation, PCGF4 in neural differentiation). Unfortunately, the novelty of this study is partially compromised by another study from Koseki's lab.

Overall, the experiments are well controlled and the conclusions are supported by the data (except the role of PCGF6 in self-renewal). I think that the paper could be improved if they dig more on the MYC-PCGF6 axis, differentiation assays and analysis of self-renewal. The current state of this manuscript is too preliminary to be published in Nature Communications.

Response: Regarding the novelty issue, we studied the role of PCGF6 in human PSCs. Distinct from the mouse report, human PSCs with PCGF6 depletion could be maintained normally, but with biased lineage differentiation capacity that favoring mesendoderm lineage and inhibiting neuroectoderm lineage. Interestingly, PCGF6 represses mesendoderm lineage differentiation via canonical PRC1-mediated WNT genes repression, and promotes neuroectoderm differentiation via noncanonical activation of SOX2 by interacting with MYC. We also identified a novel SOX2-regulatory element playing a critical role in neuroectoderm differentiation. This is the first report that dissects the function and mechanism of PCGF6 in human PSCs. Moreover, our study not only demonstrates a novel function of PCGF6 as a “switcher” of lineage-specification in human PSCs, which is different from most epigenetic factors, but also reveals the underlying mechanism of PcG proteins to regulate cell fate decision by cooperating with different factors to achieve both repressive and activated transcription.

We thank the reviewer for the positive comments on our manuscript: “Overall, the experiments are well controlled, and the conclusions are supported by the data (except the role of PCGF6 in self-renewal)”. We have also performed additional experiments to clarify the role of PCGF6 in self-renewal, as detailed below.

Major points:

1. I do not totally agree with the conclusions of the authors regarding the effect of

PCGF6 depletion in self-renewal. Although seems clear that pluripotency factors are not downregulated upon PCGF6 loss, AP stainings in figure 1 suggest that PCGF6 null cells have partially differentiated colonies. Moreover, later in the manuscript (figure 5c) they show spontaneous differentiation of PCGF6 KO cells and SOX2 SRE KO lines. Additionally, Koseki and colleagues recently showed (ref 23) a role for PCGF6 in maintaining mouse ESCs in an undifferentiated state. The authors need to perform additional analysis to clarify the role of PCGF6 in maintaining self-renewal and preventing spontaneous differentiation. For instance, among other experiments, they could determine the size of the colonies in WT and PCGF6 KO cells and the cell cycle profile of WT and KO cells.

Response: We thank the reviewer for the question and apologize for the ambiguous statement. In our study, we found the knockout of PCGF6 affects neither the pluripotent gene expression (except for SOX2) nor the maintenance of PSCs (cell growth measured by MTT assay) (Fig. 1d-1f). Therefore, we concluded that PCGF6 is unnecessary for human pluripotent stem cell maintenance.

To further confirm this conclusion, following the reviewer's suggestion, we analyzed the cell cycle and colony size of the PCGF6-knockout cells. Quantification of the colony size by Image J software revealed that PCGF6-knockout cells exhibit relatively reduced colony size as compared to wild-type cells (Fig. S1d). Cell cycle analyses with the propidium iodide (PI) staining revealed that PCGF6-knockout has little effect on cell cycle distribution (Fig. 1c). Since the MTT assay showing no significant difference in cell proliferation, these results suggest that PCGF6 knockout does not affect PSC maintenance, despite of the slightly smaller clone size.

Based on these data, we clarified the statement as “PCGF6 knockout does not affect PSC proliferation, cell cycle progression, and maintenance despite of the clone size”. This is probably because of the expression change of genes in the Rho/ROCK signal pathway (Amano et al., *Cytoskeleton* 2010) after PCGF6 deletion, shown as right.

2. Unfortunately, the connection between PCGF6 and MYC is not entirely novel: 1) PCGF6 as part of PRC1.6 which interacts with the two bona-fide MYC interacting proteins MGA and MAX, and 2), Koseki and colleagues reported co-localization of PCGF6 and MYC in mouse ESCs. Therefore, additional experiments need to be

performed to increase the novelty of the MYC-PCGF6 axis:

a. Does MYC directly interact with PCGF6?

b. Is the interaction mediated by nucleic acids?

c. Is MYC required for PCGF6 recruitment genome-wide and to the SOX2 SRE?

d. Is PCGF6 the only PRC1.6 subunit recruited to the SOX2 SRE?

e. Is PCGF6 recruited to other active enhancers in pluripotent cells occupied by MYC?

f. Is the recruitment of PCGF6 to active regulatory regions specific to MYC-occupied sites or is PCGF6 also found in enhancers occupied by other pluripotency factors?

Response: We thank the reviewer for these constructive comments. We also noticed that Dr. Koseki and his colleagues have reported the partial overlap of PCGF6-bound and MYC-bound genes in mouse ESCs (30% PCGF6-targets and 11% MYC-targets), as pointed out by the reviewer. They demonstrated that conditional ablation of *Pcgf6* in mouse ESCs leads to robust de-repression of germ cell-related genes, which in turn affects cell growth and viability. Their results show that PCGF6 recognizes the sequence-specific target by the MAX/MGA complex and mediates PRC1-dependent transcriptional silencing of germ cell-specific genes in mouse ESCs. Furthermore, their *Pcgf6*-KO mice analyses revealed that PCGF6 plays pleiotropic roles in pre- and post-implantation embryonic development. Our work here focuses on the function of PCGF6 in lineage differentiation of human PSCs and revealed that PCGF6 could function as both lineage differentiation barrier (mesendoderm) and facilitator (neuroectoderm). Moreover, our data revealed that PCGF6 achieves this lineage-biased epigenetic regulation through distinct mechanisms. Last but not least, we also found that PCGF6 could activate SOX2 transcription, revealing a novel SOX2-regulatory element and mechanism.

For the interaction between PCGF6 and MYC, we first performed co-immunoprecipitation experiment in both human PSCs (endogenous interaction) and in 293T cell with tagged PCGF6/MYC overexpression. The result showed that MYC and PCGF6 indeed interact with each other (Fig. 5e). To further determine whether the interaction is direct or depends on nucleic acids, we purified both proteins with

GST/His tag and performed *in vitro* pull-down assay. As shown in Fig. 5f, MYC and PCGF6 physically interacts with each other. This *in vitro* assay also indicated that the interaction is independent of nucleic acids or other proteins.

To determine whether MYC is required for PCGF6 recruitment to the SOX2 SRE and other targets, we constructed MYC knockdown human ESCs using two independent shRNAs and examined the binding of PCGF6 to SRE by ChIP-qPCR. Our results showed that the decreased expression of MYC results in reduced expression of SOX2 (Fig. 5h) but does not affect the enrichment of PCGF6 on SRE. We also analyzed the PCGF6 and MYC-co-occupied target genes, such as KLF1, OTX1, INPP5F and WDR36, and found that the decreased expression of MYC does not significantly change the enrichment of PCGF6 on these gene regions. Taken together, these results indicate that MYC is required for the functional activation of SOX2, but not for PCGF6 recruitment to SOX2 SRE.

To understand the binding of PCGF6 on SRE, we integrated the public available ChIP-seq data from the ENCODE/GEO database. PCGF6 is known to be in a stable complex, PRC1.6, together with RING1A/B, RYBP, L3MBTL2, E2F6, DP1, MAX, MGA, EHMT2 (also known as G9A), EHMT1 (also known as GLP), and CBX3. As shown below, we observed that some components of PRC1.6, such as MAX, E2F6 and RYBP, occupied the SRE as well. However, RNF2/RING1B, the catalytic subunit for H2AK119 ubiquitination, is not enriched on SRE. In addition, the public ChIP-seq data of HDAC2 and our ChIP-qPCR data of EHMT2 and HDAC1 all indicated that these three repressive histone modifiers are not enriched on SRE. Taken together, although PCGF6 is not the only PRC1.6 subunit recruited to the SRE sites, a functional PRC1.6 may not form on SRE sites.

For the co-occupancy of PCGF6 and MYC, we first clustered the PCGF6 binding peaks based on MYC, and then integrated with other known regulatory markers including active enhancer markers (H3K4me1, H3K27ac), active chromatin epigenetic markers (H3K4me3, P300, DNase I, CTCF), and pluripotent factors (NANOG, OCT4, SOX2). Heatmap revealed that MYC-occupied PCGF6 sites exhibited significant higher enrichment in active chromatin markers. Interestingly, these regions are also enriched with NANOG but not SOX2 or OCT4. How this kind of specificity is achieved and what the biological function is are worthy further investigation. In addition, we also clustered MYC peaks by PCGF6 binding and we did not observe significant difference between MYC-PCGF6 co-occupied sites and MYC-alone sites, in terms of H3K27ac, H3K4me3 and DNase I, Pol2 and CTCF. Thus, PCGF6 and MYC are indeed recruited to other active regulatory regions in pluripotent cells.

3. Regarding the role of PCGF6 in cell fate decisions, the authors need to perform additional assays. For instance, it is not clear if PCGF6 is important for the generation of NPC (neuroprecursors) or differentiated neurons. They authors should test both hypotheses. In line with this, mesoendoderm differentiation is still a very premature differentiation event. Is PCGF6 important for repressing cardiac, hematopoietic differentiation or endoderm cell differentiation? They should test 2 or 3 differentiation

assays to delineate more precisely the role of PCGF6 in cell fate decisions.

Response: We thank and agree with the reviewer's comments. According to the well-established protocols and our laboratory's experience, we performed lineage-directed differentiation toward pancreatic progenitors, cardiomyocytes, and neurons using wild-type and PCGF6-knockout PSCs. The result from pancreatic differentiation showed that the induction of key pancreatic lineage markers *PDX1* and *NKX6-1* is much higher in PCGF6-knockout cells (Fig. S2d and S2e). Similarly, cardiomyocytes differentiated from PCGF6-knockout human PSCs exhibits increased *T*, *NKX2.5* and *GATA4* expression (Fig. S2f and S2g). Distinct from the pancreatic and cardiac differentiation, PCGF6 knockout human PSCs showed almost no TUJ1-positive and SOX1-positive staining upon neuron differentiation, suggesting a complete block of neuron generation (Fig. S3h and S3i). Taken together, our data demonstrate that PCGF6 plays an important role in cell fate decision, favoring in neural lineage and repressing mesoderm and endoderm lineage.

S2d**S2e****S2f****S2g****S3h****S3i***Other points:**1. Please show the stability of PRC1.6 upon PCGF6 depletion.*

Response: As the reviewer suggested, we checked other subunits of PRC1.6 by western blot. The result showed that PCGF6 knockout does not affect the protein levels of other components of PRC1.6. We also performed co-immunoprecipitation with the anti-RING1B antibody in both wild-type and PCGF6-depleted human PSCs to check the integrity of PRC1.6 complex. While RING1B co-immunoprecipitates endogenous L3MBTL2, MAX, RYBP, E2F6, MGA and EHMT2 in wild-type human PSCs, the

interactions between RING1B and L3MBTL2, MAX, E2F6, MGA, EHMT2 are severely impaired in PCGF6 knockout cells. We also noticed that the depletion of PCGF6 did not disrupt the interaction between RYBP and RING1B. These results suggest that PCGF6 plays an essential role in PRC1. 6 complex integrity.

2. Figure 5e is not convincing as the SOX2 levels don't seem to decrease in the SRE KO cells

Response: To address the reviewer's concerns, we quantified the immunoblotting of SOX2. Signal intensity was normalized to the loading control α -TUBULIN. Quantification from four independent samples show that SRE knockout significantly decreases the level of SOX2 protein (around 20%-30%, comparable to PCGF6-knockout, Fig. 5e).

3. It would be important to show rescue of the neuroectoderm differentiation by

reintroducing PCGF6.

Response: Following the reviewer’s suggestion, we stably expressed PCGF6 in the PCGF6-knockout human PSCs via lentiviral transduction (PCGF6^{-/-}+PCGF6) and performed directed differentiation toward neuroectoderm. As shown in Fig. S3d-S3g, the blocking to neuroectodermal differentiation upon PCGF6 deletion is rescued by PCGF6 re-expression. This result demonstrates that the observed defective neuroectoderm phenotype is caused by PCGF6 deficiency.

Reviewer #2 (Remarks to the Author):

In this manuscript entitled “Polycomb-group ring finger 6 controls neuroectoderm specification of human pluripotent stem cells by activating SOX2 expression”, Xianchun Lan et al. aims to demonstrate that the Polycomb group ring finger 6 protein (PCGF6) controls neural-ectoderm specification of human pluripotent stem cells (PSCs) by activating SOX2 gene. The authors tested whether PCGF6 modulates pluripotency and lineage specification and contributes to early development in human cells. They tried to define the role of PCGF6 in the regulation of gene expression during early

human embryonic development by generating a PCGF6 KO in human PSCs. PCGF6 KOs maintained high levels of pluripotent markers, the ability of colony formation as well as alkaline phosphatase activity but displayed aberrant neuroectoderm differentiation during differentiation suggesting an important role as regulator of lineage specification in PSCs. By performing RNA-seq analysis in KO PSC they found 970 genes upregulated that included several mesodermal and ectodermal genes. Among the downregulated genes they found an enrichment in genes involved in nervous system development, including SOX2 which was identified as a direct target of PCGF6.

Based on these data, the authors hypothesized that PCGF6 can function as lineage switcher between mesoderm and neuroectoderm in human PSCs by both suppressing and activating mechanisms. In particular, they found that PCGF6 and MYC directly interact and co-occupy a distal regulatory element of SOX2 to activate SOX2 expression, which likely accounts for the defects in neuroectoderm differentiation.

To investigate whether the binding of PCGF6 to SOX2 was the mechanism through PCGF6 may cause destruction of the neuroectodermal differentiation, they rescued PCGF6 KO neuroectoderm defects with SOX2 ectopic expression.

The manuscript is clear and technically well organized. However, although the role of PCGF6 and PRC1.6 in regulating the cell fate decision remains interesting, the experiments fell short in supporting the proposed mechanism and its novelty in sustaining the PCGF6 role in balancing the cell fate decision.

It is well established that PCGF6 is expressed at high levels in mouse embryonic stem cells (ES), where it is required for ESC identity. The mechanism involved still remains controversial. Two reports already suggested a repressive function of Pcgf6 on mesodermal-specific and on endodermal lineage genes, while Yang et al. suggested an PRC1.6-independent direct activator function of Pcgf6 on core ESC regulators such as Oct4, Sox2 and Nanog (PMID: 25187489 PMID: 27247273). These data revealed novel roles for Pcgf6 in directly regulating Oct4, Nanog, Sox2, and Lin28 expression to maintain ESC identity. PMID: 28304275

Response: We thank the reviewer for the nice comments on our manuscript and states

that “The manuscript is clear and technically well organized”.

Previous reports have shown that *Pcgf6* is essential for mouse ESC self-renewal. *Pcgf6* knockdown decreased the expression of pluripotency markers (*Oct4*, *Nanog*, and *Sox2*) and reduced colony size in mouse ESCs (Yang et al., *Scientific reports* 2016; Zdziebło et al., *Stem cells* 2014). While *Pcgf6*-KO did not affect the levels of pluripotency markers (*Oct4*, *Nanog*, and *Sox2*) and alkaline phosphatase activity, germ cell-related genes were robustly de-repressed (Endoh et al., *eLife* 2017; Zhao et al., *Journal of biological chemistry* 2017b). Our data in human PSCs indicated that depletion of PCGF6 did not affect the expression of most pluripotent genes such as OCT4 and NANOG, consistent with previous reports on *Pcgf6*-KO mouse ESCs; however, PCGF6-knockout human PSCs maintain for many passages, exhibiting distinct features as mouse ESCs. Mechanistically, PCGF6 was reported to be enriched in OCT4 and NANOG loci in mouse ESCs (Yang et al., *Scientific reports* 2016), but in human PSCs, we did not observe the significant enrichment of PCGF6 as well as other PRC1.6 component on these two pluripotent genes. Our data further illustrated that PCGF6 is critical for proper lineage differentiation of human PSCs. Collectively, our results suggest that PCGF6 plays a different role in human PSCs from that in mouse ESCs.

The distinct function and potential regulatory mechanism of PCGF6 in mouse PSCs and human PSCs could be due to different species, or pluripotent state (i.e., naïve and primed). There are many other epigenetic enzymes exhibiting similar features. For instance, Dnmt1-deficient mouse ESCs are phenotypically normal (Tsumura et al., *Genes to cells* 2006), but undifferentiated human ESCs lacking DNMT1 are not viable (Liao et al., *Nature genetics* 2015). Disruption of TET1 does not affect the maintenance of human ESCs (González et al., *Cell stem cell* 2014), but mouse ESCs with Tet1 deletion are impaired for self-renewal (Freudenberg et al., *Nucleic acids research* 2012; Ito et al., *Nature* 2010).

It is also well known that general lack of Polycomb activity in differentiating EBs results in neuroectodermal suppression and lack of pluripotency genes expression. Thus, phenotypically, the authors are presenting a classical Polycomb phenotype reproduced

by several labs and papers in the past decades. The reduced expression of SOX2 could simply be an indirect consequence of a delayed/compromised differentiation. The overlap of both up or down regulated genes is poor and seems rather stochastic. Furthermore, the re-expression of SOX2 may simply force the system to differentiate more efficiently, resulting in a partial rescue. A similar effect may also occur upon lack of other critical Polycomb regulators.

Response: We appreciate the reviewer's comments. Our data showed that down-regulation of SOX2 expression upon PCGF6-deletion started in the undifferentiated pluripotent stage while other key pluripotent genes were not affected, indicating a specific regulation of PCGF6 on SOX2. Furthermore, when PCGF6 was re-expressed in PCGF6-KO human PSCs, the failure of neuroectodermal differentiation can be fully rescued (Fig. S3d-S3g). This result confirmed that the observed defective neuroectoderm phenotype is indeed caused by PCGF6 deficiency. In addition, when directed to differentiation toward neurons, neither TUJ1-positive (marker of neurons) nor SOX1-positive (marker of neural progenitor cell) staining was observed in PCGF6-knockout cells, indicating that the differentiation was completely blocked (Fig. S3h-S3i). Taken together, these results suggest that the neural lineage differentiation of PCGF6-deleted PSCs was severely hindered rather than delayed.

We agree with the reviewer's notion that deregulation of PcG proteins is associated with diverse phenotypes during embryonic development. For instance, *Jarid2* (a member of PRC2) -depleted ESCs are unable to differentiate into either ectodermal or mesendodermal lineage (Pasini et al., *Nature* 2010). *Pcgf3/5*-deleted ESC shows

defects in mesoderm differentiation but has no effect on endoderm and ectoderm specification (Zhao et al., *Journal of biological chemistry* 2017a). In contrast to these reports, our study here revealed that PCGF6 could function as both lineage differentiation barrier (mesendoderm) and facilitator (neuroectoderm), and PCGF6 achieves this lineage-biased regulation through distinct mechanisms.

Based on the model, one arm of PCGF6 function should be activatory together with MYC. First It's unclear what role should PCGF6 have in a complex with MYC. Such interaction was never reported before although PCGF6 has been extensively purified with proteomics approaches by several labs also in cells of neuronal lineages. This imply also that other Polycomb activities, PCGF6 co-factors as well as RING1A/B enzymatic activity should have no role in this phenotype. It is also unclear how MYC could interact with PCGF6. The model implies that this does not involve and depends on other Polycomb components, however this was not tested. Furthermore, PCGF6 forms a complex with the MGA/MAX dimer, which has affinity for E-Boxes and should be in competition with MYC DNA binding. The PCGF6 peak on SRE is not very convincing compared to snapshots on other loci and, considering that this is a bulk population of a dynamic heterogenous system, this result does not demonstrate a co-occupancy but rather represents the chromatin state of distinct cell populations suggesting a competitive behavior.

Response: Please see our response for Reviewer 1 (Question 2). Consistent with our finding, we also noticed that PCGF6 appeared in the proteins pulled-down by MYC protein in two proteome studies in HEK-293 cells and human osteosarcoma cells (Heidelberger et al., *EMBO reports* 2018; Kalkat et al., *Molecular cell* 2018).

In addition, we have knocked down MYC in human ESCs by two independent shRNAs and examined the binding of PCGF6 to SRE by ChIP-qPCR. Our results showed that the decreased expression of MYC did not affect the enrichment of PCGF6 on SRE, suggesting that MYC is not required for PCGF6 recruitment to SRE (Fig. S4i). Conversely, our data showed that the deletion of PCGF6 led to a significant reduction

of MYC binding to SRE locus. Taken together, these results indicate that PCGF6 recruits MYC to target genes and facilitates target gene activation. In addition, a very recent study reports that EZH2 directly binds cMyC via a cryptic transactivation domain, which mediates gene activation (Wang et al., *Nature cell biology* 2022). We noticed that MYC is highly expressed in both ESCs and tumor cells. Although MYC has been reported as a classic activator, it is also worth noting the multifaceted nature of the biological functions of MYC. We speculate that MYC may act as a functional antagonist of PcG complexes in a given cell environment, and competitively bind to PcG target element to regulate gene expression.

At last, the enrichment of PCGF6 on SRE was observed not only by ChIP-seq, but also confirmed by ChIP-qPCR (Fig. 5a and 5b). The results showed that PCGF6 is indeed enriched in SRE.

It is also important to highlight that it has been reported how loss of Pcgf6 in mice results in partially penetrant embryonic lethality, clearly suggesting the critical roles for PRC1.6 subunits in the early cell fate decisions (PMID: 32482889). How precisely this occurs has to be elucidated but no links with the proposed in vitro phenotype are available. Overall, although the correlative gene ontology analyses goes in the direction of the proposed model, the manuscript remains incremental its first part and lacks several mechanistic information to support the proposed model.

Response: Disruption of PCGF6 has distinct effects on human ESCs and mouse ESCs. In previous reports, Pcgf6 was identified as an essential self-renewal gene in mouse ESCs regulating pluripotent and germ cell-specific genes. Our PCGF6-knockout human PSC lines showed that PCGF6 is not required for the maintenance of human PSCs. The different effects of PCGF6 on mouse PSCs and human PSCs might be contributed by species differences, or pluripotent state (i.e., naïve and primed). It is well-established that mouse and human ESCs represent different pluripotency stages. Previous studies have showed that PCGF6-KO mice are viable but less numbers than the Mendelian ratio, and the surviving homozygotes exhibited growth retardation, suggesting a subtle role for PCGF6 in embryonic development. Embryonic death in the PCGF6-KO mice could be continuously detected during post-implantation development (Endoh et al., *eLife* 2017; Liu et al., *Journal of biological chemistry* 2020). Although PCGF6 is dispensable for the maintenance of human PSCs, our data indicated that PCGF6 is essential for normal development, consistent with the abnormalities in later development. Interestingly, DNMT1, a classical epigenetic enzyme, has similar phenotype between humans ESC and mice ESC. In human ESCs, disruption of DNMT1

showed rapid cell death (Liao et al., *Nature genetics* 2015). However, DNMT1-deficient mouse ESCs are phenotypically normal but died upon the induction of cellular differentiation (Tsumura et al., *Genes to cells* 2006). While disruption of TET1 does not affect the maintenance of human ESCs (González et al., *Cell stem cell* 2014), mouse ESCs with Tet1 deletion exhibits impaired self-renewal (Freudenberg et al., *Nucleic acids research* 2012; Ito et al., *Nature* 2010). So, similar to DNMT1 and TET1, PCGF6 has distinct functions in human and mouse ESCs. It will be interesting to explore the effect of PCGF6 deletion under human ‘naive’ conditions, which may help identify the stage of PCGF6-dependent pluripotency.

Mechanistically, our data demonstrate that PCGF6 acts as a transcriptional repressor for genes involved in the WNT signaling pathway through PRC1.6-mediated repression. This helps explain the phenotype that loss of PCGF6 leads to skewed differentiation into the mesendoderm lineage. We also found that PCGF6 directly activates the expression of SOX2 via binding to a regulatory element of SOX2, which contributes to neuroectoderm differentiation. Moreover, we identify MYC as a functional co-factor of PCGF6 and co-activates the expression of SOX2 with PCGF6 in our model. Overall, our finding provides a novel insight that PcG proteins may cooperate with specific transcription factors and play distinct roles in the regulation of different lineage differentiations, not limiting to its canonical repressor functions.

Reviewer #3 (Remarks to the Author):

In this manuscript, Lan et al. investigated the function of PCGF6 in human PSCs maintenance and differentiation. The results showed that genetic deletion of PCGF6 did not affect the PSC self-renewal but favored the differentiation of PSCs towards the mesendodermal lineage. Genome-wide transcriptome and epigenome analysis showed that PCGF6, RING1B, RYBP bound to the promoters of Wnt signaling target genes and repressed their expression. In contrast, PCGF6 activated the Sox2 gene expression by binding to its enhancer regions with MYC. Deletion of Sox2-regulatory element

phenocopied the impaired neuroectoderm differentiation of PCGF6-deleted PSCs. The authors proposed that PCGF6 can function as lineage switcher between mesendoderm and neuroectoderm through either suppression or activation mechanisms.

Overall, the results in this MS are well presented and convincing. However, several issues need to be addressed:

Response: We thank the reviewer for the positive comment stating that “Overall, the results in this MS are well presented and convincing”. We have addressed the specific comments below.

Major points:

1. Different from mouse ESCs in which the PRC1.6-repressed targets are enriched with meiosis and germ cell-related genes, deletion of PCGF6 in human PSCs de-repressed much broader genes involving mesendodermal lineage development (Fig. 2A). It is unclear whether these genes are specifically repressed by PRC1.6 or by overlapped effects from other nPRC1 variants because (1) the ChIP-seq results in Figure 3e did not show that genome-wide PRC1.6-specific components such as E2F6 and Max/MGA co-occupied with PCGF6 and the data was only shown at two gene loci (Fig. 3g); (2) RING1B and RYBP are not PRC1.6-specific components; (3) in the PCGF6-KO cells, the uH2A signals was reduced but not completely lost, suggesting other PRC1-complexes still bound to chromatin and mediated uH2A. The authors should perform the ChIP-seq analysis using PRC1.6-specific components such as E2F6 and MAX/MGA in both wild-type and PCGF6-knockout cells to define the genes that are specifically regulated by the PCGF6-containing PRC1.6 complex before drawing the conclusion that PRC1.6 represses the mesoendodermal lineage-specific genes in human PSCs.

Response: We appreciate the reviewer’s comments and have performed the experiments suggested by the reviewer. We first performed ChIP-seq using the anti-MAX and anti-E2F6 antibody in wild-type and PCGF6-depleted human PSCs. Data analyses revealed that MAX and E2F6 are enriched in many genes of WNT signaling

pathway bound by PCGF6. Venn diagram shows that there are 398 genes that are directly repressed by PCGF6 and targeted by PRC1.6 (overlapped with targets of RING1B, RYBP, E2F6 and MAX), and many genes are within the WNT signaling pathway (such as FZD9, WIN5A, WIN11). However, the signal of the MGA and E2F6 peaks in PCGF6-KO group are almost unchanged or even slightly increased, as shown in the heatmap below. This observation is consistent with a previous report (Stielow et al., *PLoS genetics* 2018). We think this may be due to the ability of MAX and E2F6 to bind DNA while PCGF6 cannot. PCGF6 may act as a bridge to maintain the integrity of PRC1.6. To test this notion, we investigated whether the components of PRC1.6 such as RNF2, HDAC1 and HDAC2, are also enriched for genes in WNT signaling pathway after PCGF6 knockout. ChIP-qPCR results showed that the enrichment of RNF2, EHMT2 and HDAC1 on genes in the WNT signaling pathway is indeed significantly reduced in the absence of PCGF6.

2. The finding that PCGF6 and MYC co-occupied at the Sox2 enhancer was novel, and the ChIP-seq qPCR results suggested that PCGF6 recruited MYC to the Sox2 enhancer. It was unclear whether this was a unique mechanism at the Sox2 locus. Would it be nice if the authors could do some analysis to examine whether the mechanisms found at the Sox2 locus are also utilized to regulate the 237 downregulated genes in the PCGF6-KO cells (Fig. 5h).

Response: We appreciate the reviewer's comments. To determine whether the mechanisms operated at the SRE locus are also used to regulate the overlapping down-regulated genes, we performed MYC-ChIP-seq assay in wild-type and PCGF6-KO human PSCs. In this dataset, we identified 80 (80/302) downregulated targets in PCGF6-KO cells that are with MYC occupancy (the number based on MYC ChIP-seq data in ENCODE was 237). Importantly, our data showed that loss of PCGF6

significantly reduced the enrichments of MYC on the 80 downregulated genes (Fig. S4g and S4h). Moreover, interrogation of additional chromatin marks or transcription factors revealed that a large majority of PCGF6-MYC co-binding sites overlaps with gene-activation-related H3K27ac, H3K4me3, P300, DNase1, POL2 and CTCF peaks. Thus, PCGF6-MYC-shared binding sites exist in human PSCs and are potentially involved in transcriptional activation not only at the SRE locus, but also at the 80 downregulated genes after PCGF6 KO.

3. It is unclear what the mechanisms determine the recruitment of PRC1.6 or PCGF6-MYC at different genomic loci shown in the model (Fig.6). Would it be nice if the authors include some discussion in the discussion section?

Response: We thank the reviewer for bringing up this question.

The mechanisms underlying the recruitment of PRC1.6 or PCGF6-MYC to target sites may be complicated. By performing biochemical and bioinformatics analyses, we

discovered that PCGF6-activated genes prefer to a more open chromatin conformation, which may allow transcription activators or coactivators (such as P300, DNase1, POL2 and CTCF) to access. Therefore, there might be a competitive model, wherein PCGF6, MYC and other partners, through multiple cooperating interactions, compete for the target element of PcG proteins during embryogenesis.

First, our data revealed that deletion of PCGF6 led to a significant reduction of MYC binding to SRE locus (Fig. 5g), suggesting that PCGF6 is necessary for the recruitment of MYC to SRE. To test whether MYC is also required for PCGF6 binding to SRE, we constructed MYC-knockdown human ESCs using two independent shRNAs and examined the binding of PCGF6 to SRE via ChIP-qPCR. Our new data showed that the decreased expression of MYC did not affect the enrichment of PCGF6 on SRE locus (Fig. S4i), indicating that MYC is not essential for PCGF6 recruitment to the SRE locus.

Furthermore, by integrating the ChIP-seq data of human ESCs, we discovered a subset of PCGF6 peaks that are overwhelmingly enriched for MYC but have a relatively low overlap with H2AK119ub (41%). This group of genes are termed as PCGF6-MYC-regulated sites (Cluster 1). Another subset of PCGF6 sites overlapped with H2AK119ub significantly (36%), which represents canonical PRC1.6 sites (Cluster 2). In cluster 1, PCGF6 cooperates with MYC, and these genes have relatively lower levels of H2AK119ub and RING1B binding. In cluster 2, the genes have higher levels of

H3K27me3 compared to cluster 1. We also noticed that cluster 1 is not only specific to MYC but also occupied by the gene-activation-related H3K27ac, H3K4me3 and P300 peaks, contrast to cluster 2. PCGF6-MYC-shared peaks prefer to a more open conformation chromatin compared with canonical PRC1.6 regions, which may allow transcriptional activators to access. These (co)activators might cooperatively contribute to the co-targeting of PCGF6 and MYC to PCGF6-MYC-shared sites where gene-activation-related markers coexist.

A very recent study reports that EZH2 directly binds to cMyc *via* a cryptic transactivation domain, which mediates gene activation (Wang et al., *Nature cell biology* 2022). They proposed a “coalition model”: multiple cooperating interactions exist among cMyc and many partners, possibly through the phase-separated condensates formed by the transactivation domain and/or the unstructured protein region of EZH2, cMyc and (co)activators. We noticed that MYC is highly expressed in

both ESCs and tumor cells. Although MYC has been reported as a classic activator, it is also worth noting the multifaceted nature of the biological functions of MYC in addition to the most-studied role in transcriptional regulation. Thus, MYC may act as a functional antagonist of PcG complexes in a given situation, and competitively bind to PcG target element to regulate gene expression.

Minor points:

1. Fig. 3e: RING1B signal is very low. The plots should also include the input control.

Response: We have added the INPUT control data in Fig. 3e.

2. All photos missed the scale bars.

Response: We have now included the scale bars in images.

References:

- Amano, M., Nakayama, M., and Kaibuchi, K. (2010). Rho-kinase/ROCK: A key regulator of the cytoskeleton and cell polarity. *Cytoskeleton (Hoboken, NJ)* 67, 545-554.
- Endoh, M., Endo, T.A., Shinga, J., Hayashi, K., Farcas, A., Ma, K.W., Ito, S., Sharif, J., Endoh, T., Onaga, N., et al. (2017). PCGF6-PRC1 suppresses premature differentiation of mouse embryonic stem cells by regulating germ cell-related genes. *eLife* 6.
- Freudenberg, J.M., Ghosh, S., Lackford, B.L., Yellaboina, S., Zheng, X., Li, R., Cuddapah, S., Wade, P.A., Hu, G., and Jothi, R. (2012). Acute depletion of Tet1-dependent 5-hydroxymethylcytosine levels impairs LIF/Stat3 signaling and results in loss of embryonic stem cell identity. *Nucleic acids research* 40, 3364-3377.
- González, F., Zhu, Z., Shi, Z.D., Lelli, K., Verma, N., Li, Q.V., and Huangfu, D. (2014). An iCRISPR platform for rapid, multiplexable, and inducible genome editing in human pluripotent stem cells. *Cell stem cell* 15, 215-226.
- Heidelberger, J.B., Voigt, A., Borisova, M.E., Petrosino, G., Ruf, S., Wagner, S.A., and Beli, P. (2018). Proteomic profiling of VCP substrates links VCP to K6-linked ubiquitylation and c-Myc function. *EMBO reports* 19.
- Ito, S., D'Alessio, A.C., Taranova, O.V., Hong, K., Sowers, L.C., and Zhang, Y. (2010). Role of Tet proteins in 5mC to 5hmC conversion, ES-cell self-renewal and inner cell mass specification. *Nature* 466, 1129-1133.
- Kalkat, M., Resetca, D., Lourenco, C., Chan, P.K., Wei, Y., Shiah, Y.J., Vitkin, N., Tong, Y., Sunnerhagen, M., Done, S.J., et al. (2018). MYC Protein Interactome Profiling Reveals Functionally Distinct Regions that Cooperate to Drive Tumorigenesis. *Molecular cell* 72, 836-848.e837.
- Liao, J., Karnik, R., Gu, H., Ziller, M.J., Clement, K., Tsankov, A.M., Akopian, V., Gifford, C.A., Donaghey, J., Galonska, C., et al. (2015). Targeted disruption of DNMT1, DNMT3A and DNMT3B in human embryonic stem cells. *Nature genetics* 47, 469-478.
- Liu, M., Zhu, Y., Xing, F., Liu, S., Xia, Y., Jiang, Q., and Qin, J. (2020). The polycomb group protein PCGF6 mediates germline gene silencing by recruiting histone-modifying proteins to target gene promoters. *The Journal of biological chemistry* 295,

9712-9724.

Pasini, D., Cloos, P.A., Walfridsson, J., Olsson, L., Bukowski, J.P., Johansen, J.V., Bak, M., Tommerup, N., Rappsilber, J., and Helin, K. (2010). JARID2 regulates binding of the Polycomb repressive complex 2 to target genes in ES cells. *Nature* 464, 306-310.

Stielow, B., Finkernagel, F., Stiewe, T., and Nist, A. (2018). MGA, L3MBTL2 and E2F6 determine genomic binding of the non-canonical Polycomb repressive complex PRC1.6. *PLoS genetics* 14, e1007193.

Tsumura, A., Hayakawa, T., Kumaki, Y., Takebayashi, S., Sakaue, M., Matsuoka, C., Shimotohno, K., Ishikawa, F., Li, E., Ueda, H.R., et al. (2006). Maintenance of self-renewal ability of mouse embryonic stem cells in the absence of DNA methyltransferases Dnmt1, Dnmt3a and Dnmt3b. *Genes to cells: devoted to molecular & cellular mechanisms* 11, 805-814.

Wang, J., Yu, X., Gong, W., Liu, X., Park, K.S., Ma, A., Tsai, Y.H., Shen, Y., Onikubo, T., Pi, W.C., et al. (2022). EZH2 noncanonically binds cMyc and p300 through a cryptic transactivation domain to mediate gene activation and promote oncogenesis. *Nature cell biology* 24, 384-399.

Yang, C.S., Chang, K.Y., Dang, J., and Rana, T.M. (2016). Polycomb Group Protein Pcgf6 Acts as a Master Regulator to Maintain Embryonic Stem Cell Identity. *Scientific reports* 6, 26899.

Zdzieblo, D., Li, X., Lin, Q., Zenke, M., Illich, D.J., Becker, M., and Muller, A.M. (2014). Pcgf6, a polycomb group protein, regulates mesodermal lineage differentiation in murine ESCs and functions in iPS reprogramming. *Stem cells (Dayton, Ohio)* 32, 3112-3125.

Zhao, W., Huang, Y., Zhang, J., Liu, M., Ji, H., Wang, C., Cao, N., Li, C., Xia, Y., Jiang, Q., et al. (2017a). Polycomb group RING finger proteins 3/5 activate transcription via an interaction with the pluripotency factor Tex10 in embryonic stem cells. *The Journal of biological chemistry* 292, 21527-21537.

Zhao, W., Tong, H., Huang, Y., Yan, Y., Teng, H., Xia, Y., Jiang, Q., and Qin, J. (2017b). Essential Role for Polycomb Group Protein Pcgf6 in Embryonic Stem Cell Maintenance and a Noncanonical Polycomb Repressive Complex 1 (PRC1) Integrity.

The Journal of biological chemistry 292, 2773-2784.

REVIEWER COMMENTS

Reviewer #1 (Remarks to the Author):

Although the authors have addressed most of my concerns, there are still few things that need to be clarified.

I think that the pictures and AP stainings from figure 1b should be improved. Both show differentiated cells, which is in contrast with all the other results from this figure. Also, the PCGF6 KO colonies are not "relatively smaller", according to the quantification in figure s1d they should be significantly smaller. Please perform statistical analysis.

I must admit that it was difficult to follow the point-by-point letter because some figures are not incorporated in the new version of the manuscript. For instance, PCGF6 ChIPs in page 7 and the heatmaps in page 9 of the point-by-point letter, are not included in the manuscript. Please include all these figures in the revised version.

Regarding the SOX2 protein levels in the PCGF6 KO. I am still not convinced that they are decreased and therefore should be further investigated. They could titrate the protein lysate and perform WB.

In mESC, RING1B is recruited to the SRE. To conclude that RING1B is not recruited to the human SRE, the authors should check other RING1B datasets or perform a RING1B ChIP-qPCR. Otherwise, the role of PCGF6 in regulating SOX2 would be PRC1 independent. This is quite an important point.

Reviewer #3 (Remarks to the Author):

The authors have addressed most of my comments on the original submission and the revision is an improvement in my view. I recommend to publish the manuscript on NC.

1 **Response to the comments from Reviewer #1**

2

3

4 *Although the authors have addressed most of my concerns, there are still few things*
5 *that need to be clarified.*

6 *I think that the pictures and AP stainings from figure 1b should be improved. Both*
7 *show differentiated cells, which is in contrast with all the other results from this figure.*

8 *Also, the PCGF6 KO colonies are not “relatively smaller”, according to the*
9 *quantification in figure s1d they should be significantly smaller. Please perform*
10 *statistical analysis.*

11 **Response:** We are sorry for that the figures and interpretation of our results were
12 unclear. We have now provided images in Figure 1b with higher resolution. We
13 examined more phase contrast and AP staining pictures with different field sizes.
14 These results suggest that although the colonies of PCGF6-KO human PSCs are
15 generally flat and loose and a few even look like differentiated, these cells still display
16 positive alkaline phosphatase staining and mostly OCT4-positive. Combined with
17 other results (cell cycle and proliferation analysis, immunostaining), we thus consider
18 that PCGF6-KO human PSCs display morphologically distinctive but without
19 inducing significant differentiation.

20

21

22 In addition, we accordingly performed statistical analysis, which suggests that the
 23 PCGF6 KO colonies are significantly smaller (Figure S1d). Therefore, we reworded
 24 the sentence as follows: “PCGF6 knockout did not affect the iPSC cell cycle,
 25 proliferation and maintenance although the colony size became significantly smaller
 26 (Figure 1c-1d and Figure S1d).”

S1d

27

28

29 *I must admit that it was difficult to follow the point-by-point letter because some*
 30 *figures are not incorporated in the new version of the manuscript. For instance,*
 31 *PCGF6 ChIPs in page 7 and the heatmaps in page 9 of the point-by-point letter, are*
 32 *no included in the manuscript. Please include all these figures in the revised version.*

33 **Response:** We have now included these figures in the re-revised version, such as
 34 Figure S2i-S2k (analyses of PRC1.6 ChIP-seq and ChIP-qPCR at WNT gene loci),
 35 Figure S4a-S4c (analyses of PRC1.6 ChIP-seq and ChIP-qPCR at SRE locus), Figure

36 S4g (immunoprecipitation experiment in 293T cell with tagged PCGF6/MYC
37 transfection), Figure S4k (analyses of PCGF6 ChIP-qPCR at PCGF6- and
38 MYC-co-occupied target genes after MYC knockdown), Figure S5d-S5e (analyses of
39 the co-occupancy of PCGF6 and MYC).

40

41 *Regarding the SOX2 protein levels in the PCGF6 KO. I am still not convinced that*
42 *they are decreased and therefore should be further investigated. They could titrate the*
43 *protein lysate and perform WB.*

44 **Response:** We thank the reviewer for this nice suggestion. To further confirm this
45 conclusion, we have performed titrated western blot following the reviewer's
46 suggestion. The result show that PCGF6 depletion caused significant decrease of
47 SOX2 expression (about 50%).

48

49

50 *In mESC, RING1B is recruited to the SRE. To conclude that RING1B is not recruited*
51 *to the human SRE, the authors should check other RING1B datasets or perform a*
52 *RING1B ChIP-qPCR. Otherwise, the role of PCGF6 in regulating SOX2 would be*
53 *PRC1 independent. This is quite an important point.*

54 **Response:** We thank the reviewer for raising this suggestion. We first analyzed a

55 number of public RING1B ChIP-seq datasets in human and mouse separately, and the
 56 results showed that RING1B is enriched in *Sox2* locus in mouse ESCs but not in
 57 human ESCs or other human cell lines. Furthermore, we performed RING1B
 58 ChIP-qPCR and the result indicated that RING1B is not recruited to the SRE in
 59 human ESCs. Thus, combined with our previous results, we consider that the role of
 60 PCGF6 in regulating SOX2 in human PSCs might be independent of PRC1.6 (neither
 61 strong H2Aub nor significant RING1B binding) but related to MYC.

62
 63 (I ~ II : PCGF6 ChIP-seq in mouse ESCs (GSE84905); III~IX: RING1B ChIP -seq
 64 in different mouse ESCs (GSE161808, GSE162739, GSE157748))

65

66

67 (Top panel: SRE loci. Bottom panel: positive control. **1**: PCGF6 ChIP-seq in this
68 study; **2**: H2AK119Ub ChIP-seq in this study; **3~12**: RING1B ChIP-seq in human cell
69 lines, including **3**: H1 cell line (GSM2805868); **4**: HUES64 cell line (GSM2805868);
70 **5**: WA09/H9 reprogramed naïve human PSCs (GSE164786); **6**: WA09/H9 primed
71 human PSCs (GSE164786); **7**: Bone-marrow-derived primary human mesenchymal
72 stem cells (hMSC) (GSE125166); **8**: osteogenic cells derived from
73 bone-marrow-derived primary human mesenchymal stem cells (hBMSC)
74 (GSE125166); **9**: ME-1 cell line (GSE128771); **10**: HEK 293FT cells (GSE175673);
75 **11**: the human SCLC cell line (NCI-H1963, GSE191106); **12**: human chronic myeloid
76 leukemia cell line (K562, GSE167869))
77

78

REVIEWERS' COMMENTS

Reviewer #1 (Remarks to the Author):

The authors have addressed my concerns in a thoughtful manner. I have no further comments.